# DiffNat: Improving diffusion image quality using natural image statistics

## Abstract

Diffusion models have advanced generative AI significantly in terms of editing and creating naturalistic images. However, efficiently improving generated image quality is still of paramount interest. In this context, we propose a generic "naturalness" preserving loss function, viz., kurtosis concentration (KC) loss, which can be readily applied to any standard diffusion model pipeline to elevate the image quality. Our motivation stems from the projected kurtosis concentration property of natural images, which states that natural images have nearly constant kurtosis values across different band-pass versions of the image. To retain the "naturalness" of the generated images, we enforce reducing the gap between the highest and lowest kurtosis values across the band-pass versions (e.g., Discrete Wavelet Transform (DWT)) of images. Note that our approach does not require any additional guidance like classifier or classifier-free guidance to improve the image quality. We validate the proposed approach for three diverse tasks, viz., (1) personalized few-shot finetuning using text guidance, (2) unconditional image generation, and (3) image super-resolution. Integrating the proposed KC loss has improved the perceptual quality across all these tasks in terms of both FID, MUSIQ score, and user evaluation.

## 1 Introduction

Multi-modal generative AI has advanced by leaps and bounds with the advent of the diffusion model. Large-scale text-to-image diffusion models, e.g., DALLE Ramesh et al. (2022), Stable-diffusion Rombach et al. (2022) etc. synthesize high-quality images in diverse scenes, views, and lighting conditions from text prompts. The quality and diversity of these generated images are astonishing since they have been trained on a large collection of image-text pairs and are able to capture the visual-semantic correspondence effectively. Although the diffusion model-generated images look realistic, a recent study has shown that the generated images can be distinguished from natural images using state-of-the-art image forensic tools Corvi et al. (2023). This implies that state-of-the-art generative models might be good at image editing, but often leave unnatural traces and lack "naturalness" quality. This problem is more prevalent in the cases of few-shot finetuning of large multi-modal diffusion models, e.g., "personalization" of text-to-image diffusion model. Popular methods, e.g., DreamBooth Ruiz et al. (2022), Custom diffusion Kumari et al. (2022), etc. achieve impressive subject-driven "personalized" image generation based on text prompts, but these have several limitations, e.g., image quality degradation due to unnatural artifacts, etc. Image quality is of utmost importance for other generative tasks as well, e.g., super-resolution, image restoration, unconditional image generation, etc. Some examples of unnatural artifacts are shown in Fig. 1.

To improve image quality, several methods rely on guidance methods, e.g., classifier guidance, and classifier-free guidance Dhariwal & Nichol (2021) etc. However, these methods require external supervision and add complexity to the training process. Our goal is to improve the image quality without any additional guidance, yet preserving the "naturalness" of the generated images by exploring the well-known kurtosis concentration property of natural images Zhang & Lyu (2014). This property states that natural images have nearly constant kurtosis (fourth order moment) values across different band-pass (e.g., Discrete Cosine Transform (DCT), Discrete Wavelet Transform (DWT)) versions of the images Zhang & Lyu (2014). Inspired by this property, we propose a novel kurtosis concentration (KC) loss, which is generic and applicable to any diffusion based pipeline. More specifically, this

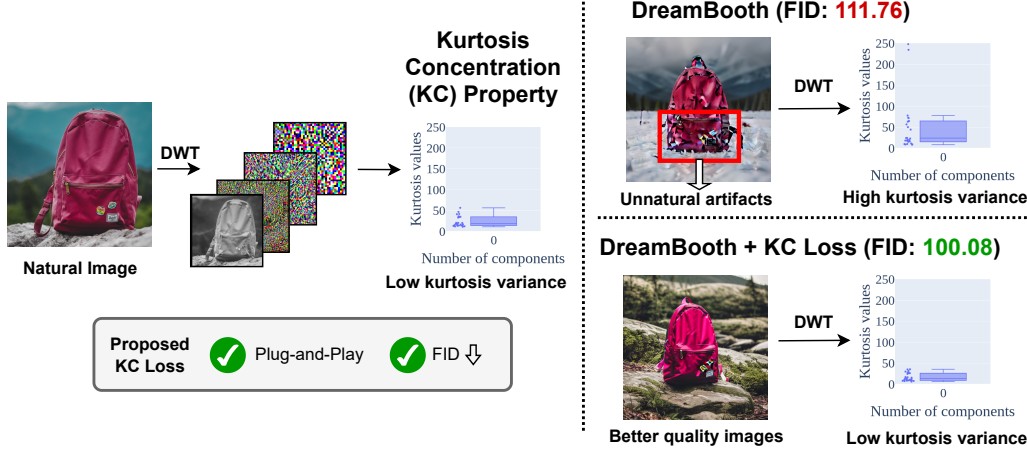

Figure 1: Overview of DiffNat. We utilize the kurtosis concentration (KC) property of natural images, which states the kurtosis values across different bandpass filtered (Discrete Wavelet Transform (DWT)) version of the images tend to be constant (left). As can be observed in this figure, 50 percentile of the kurtosis values reside in the blue box, which indicates the concentration of the kurtosis values. For natural images, this spread is relatively smaller. Inspired by this observation, we propose a novel KC loss, which minimize the deviation of kurtosis across different bandpass (DWT) versions of images. This loss can be applied to any diffusion pipleline with the traditional reconstruction loss. Here, we consider DreamBooth Ruiz et al. (2022). DreamBooth generated images might have unnatural artifacts, producing high kurtosis variance (large spread in the blue box) and higher FID (right top). The FID over the DreamBooth dataset is reported here. Adding KC loss improves image quality in terms of FID and reduces the kurtosis variance (right bottom).

loss minimizes the gap in the kurtosis of an image across band-pass filtered versions and thus enforce the "naturalness" of the generated images.

This loss is general-purpose and does not even require any labels. It can be adapted to various generative tasks with minimal effort. In this work, we experiment with diverse tasks of: (1) personalized few-shot finetuning of text-to-image diffusion model, (2) unconditional image generation, and (3) image super-resolution.

Our major contributions are as follows:

- We introduce DiffNat - a framework for improving the image quality of diffusion models using natural image statistics. Based on the kurtosis concentration property, we propose a novel loss function by minimizing the gap of kurtosis values (i.e., the difference between maximum and minimum kurtosis values) across the band-pass (in DWT domain) filtered version of the image. To the best of our knowledge, we are the first to propose this loss based on natural image statistics.
- We provide theoretical insights into the proposed loss function for generating images with better perceptual quality.
- We validate the proposed loss in diverse generative tasks, e.g., (1) personalized few-shot finetuning of text-to-image diffusion model using text guidance, (2) unconditional image generation, and (3) image super-resolution. Experiments suggest that incorporating the proposed loss improves the perceptual quality in all these tasks across different benchmarks. We have validated the proposed approach with a user study as well.

## 2    RELATED WORK

**Deep Generative Models.** Generative models (GANs Goodfellow et al. (2020), VAEs Kingma et al. (2019), flow-based models Rezende & Mohamed (2015), and diffusion models Ho et al. (2020)) learn the probability distribution of given data, allowing us to sample new data points from the distribution. Deep generative models have been used for modeling the distribution of faces Karras et al. (2019), 3D objects Wu et al. (2016), videos by Vondrick et al. (2016), natural images by Karras et al. (2019); Brock et al. (2018), etc for unconditional synthesis. Conditioning the generative models on segmentation mask Isola et al. (2017), class label Mirza & Osindero (2014), text Tao et al. (2022) enables us to have more control over the generated images. Generative models can be controlled

using guidance from images, texts, etc. ILVR Choi et al. (2021) present an iterative way to guide the image synthesis process using a reference image. Instance-conditioned GAN Casanova et al. (2021) allows for generating semantic variations of a given reference image, by training using the nearest neighbors of the reference image. Roich et al. (2022) fine-tune the generator around an inverted latent code anchor, allowing for latent-based semantic editing on images that are out of the generator's domain.

**Text-to-Image Generation and Editing.** Generating high fidelity, diverse images using text inputs has seen tremendous progress recently. Many approaches based on GANs have been proposed for text-to-image generation Qiao et al. (2019); Tao et al. (2022); Liao et al. (2022); Zhu et al. (2019); Ruan et al. (2021). More recent advances in text-based image synthesis (Stable Diffusion Rombach et al. (2022), Imagen Saharia et al. (2022), etc) have been powered by diffusion models trained on massive datasets. GAN-based text-based image editing approaches Crowson et al. (2022); Bau et al. (2021); Abdal et al. (2022); Gal et al. (2021); Patashnik et al. (2021) have made significant strides recently thanks to CLIP Radford et al. (2021). Diffusion-based text-to-image editing methods Ruiz et al. (2022); Kumari et al. (2022); Gal et al. (2022) show better control and impressive editing results. For "personalizing" these text-to-image models, Textual Inversion Gal et al. (2022) represent a subject as a new "word" in the embedding space of a diffusion model, which is used in natural language prompts for creating new images of the subject in novel scenes. DreamBooth Ruiz et al. (2022) embeds the subject in the output domain of the model and the resulting unique identifier is used to synthesize novel images of the subject in unseen contexts. Custom Diffusion Kumari et al. (2022) extends this by enabling the composition of multiple new concepts with existing ones.

**Natural Image Statistics.** Natural images have interesting scale-invariance and noise properties Zoran & Weiss (2009), which has been used for image restoration problems. Projected kurtosis concentration property of natural images, i.e., natural images tend to have constant kurtosis values across different band-pass (DCT, DWT) filtered version has been used for blind forgery detection Zhang & Lyu (2014).

## 3 METHOD

In this section, we present the concept of the kurtosis concentration loss, which can be applied to various generative tasks for enhancing the quality of generated images. We start with providing a basic understanding of the kurtosis concentration property of natural images and how we leverage this property to propose kurtosis concentration (KC) loss which enforces the "naturalness" of the generated samples.

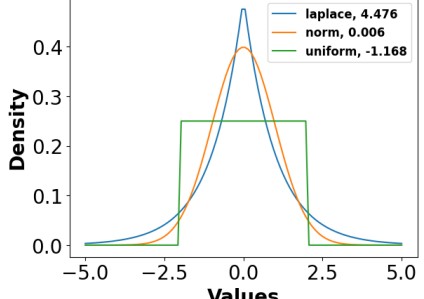

### 3.1 KURTOSIS CONCENTRATION PROPERTY

**Definition 1** *Kurtosis : Kurtosis is a measure of the "peakedness" of the probability distribution of a random variable Zhang & Lyu (2014). For a random variable $x$, its kurtosis is defined as,*

Figure 2: Kurtosis of various distributions. Intuitively, kurtosis captures the peakedness of the distribution.

$$\kappa(x) = \frac{\mu_4(x)}{(\sigma^2(x))^2} - 3. \qquad (1)$$

*where $\sigma^2(x) = \mathbb{E}_x[(x - \mathbb{E}_x(x))^2]$ and $\mu_4(x) = \mathbb{E}_x[(x - \mathbb{E}_x(x))^4]$ is the second order and fourth order moment of $x$. E.g., Gaussian random variable has kurtosis value 0.*

Intuitively, kurtosis is a measure of the peakedness of a distribution. Kurtosis of well-known distributions is shown in Fig. 2. A positive kurtosis indicates that the distribution is more peaked than the normal distribution and negative kurtosis indicates it to be less peaked than normal distribution Zhang & Lyu (2014). Kurtosis is a useful statistic used for blind source separation Naik et al. (2014) and independent component analysis (ICA) Stone (2002).

For a random vector $x$, we define the kurtosis of the 1D projection of $x$ onto a unit vector $w$ as projection kurtosis, i.e., $\kappa(w^T x)$. This projection kurtosis is an effective measure for the statistical properties of high-dimensional data. E.g., if $x$ is a Gaussian, its projection over any $w$ has a 1D

Gaussian distribution. Therefore, its projection kurtosis is always zero, which exhibits the kurtosis concentration (to a single value, i.e., zero) of Gaussian.

It is well-known that natural images can be modeled using zero-mean GSM vector Zoran & Weiss (2009). Next, we analyze an interesting property of the GSM vector.

**Lemma 1** *A Gaussian scale mixture (GSM) vector x with zero mean has the following probability density function:*

$$p(x) = \int_0^\infty \mathcal{N}(x; 0, z\Sigma_x) p_z(z) dz \tag{2}$$

*and its projection kurtosis is* *constant* *with respect to the projection direction w, i.e.,*

$$\kappa(w^T x) = \frac{3 var_z\{z\}}{\mathcal{E}_z\{z\}^2} \tag{3}$$

*where $\mathcal{E}_z\{z\}$ and $var_z\{z\}$ are the mean and variance of latent variable z respectively.*

*Proof.* The proof is provided in the supplementary material.

This result by Zhang & Lyu (2014) shows that projection kurtosis is constant across projection directions (e.g., wavelet basis), which provides a theoretical understanding of the kurtosis concentration property, which we will discuss next.

**Kurtosis Concentration Property**: It has been observed that for natural images, kurtosis values across different band-pass filter channels tend to be close to a constant value. This is termed as kurtosis concentration property of natural images Zhang & Lyu (2014); Zoran & Weiss (2009). It can be interpreted as an implication of Lemma 1, if we consider natural images as zero-mean GSM vector. As a motivating example, we demonstrate the kurtosis concentration property of natural images in Fig. 1. Next, we establish the relation between the projection kurtosis of the noisy version of the image and the corresponding signal-to-noise ratio.

**Lemma 2** *If the noisy version of the natural image is denoted by, y = x + n, where x is a whitened GSM vector (normalized natural image) and n is a zero-mean white Gaussian noise with variance $\sigma^2 I$, x and n are mutually independent of each other, then the projection kurtosis of y, $\kappa(w^T y)$ can be expressed as:*

$$\kappa(w^T y) = \kappa(w^T x)\left(1 - \frac{c}{SNR(y)}\right)^2 = \frac{3 var_z\{z\}}{\mathcal{E}_z\{z\}^2}\left(1 - \frac{c}{SNR(y)}\right)^2 \tag{4}$$

*where Signal-to-Noise Ratio (SNR) is defined as, $SNR(y) = \frac{\sigma^2(y)}{\sigma^2(n)}$ and c is a constant.*

*Proof.* The proof is provided in the supplementary material.

This result utilizes the fact that, natural images have constant projection kurtosis, stated in Lemma 1. Next, we connect projection kurtosis minimization to denoising.

**Proposition 1** *Minimizing projection kurtosis denoise input signal.*

From Lemma 2, we can observe there exists an inverse relation between the projection kurtosis and image quality (SNR), therefore minimizing projection kurtosis will increase SNR and the image will be denoised better.

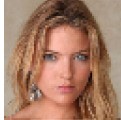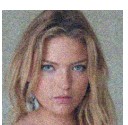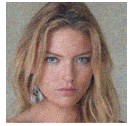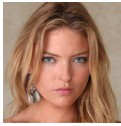

Input     GD (SNR =18.2)     GD + KC (SNR=20.4)     GT

Figure 3: Empirical evidence of proposition 1, i.e., minimizing KC loss denoise input signal. Here we take standard guided diffusion (GD) model with and without kurtosis loss for inference for 400 steps, and the denoised outputs are shown. The model trained with KC loss generates better quality images, which is also reflected in higher SNR values. GT refers to ground truth.

The primary objective of diffusion models is to learn denoising from a noisy image or latent embedding in order to generate a clean image. Then by Lemma 2, the projection kurtosis minimization results in better denoising (high SNR) of the reconstructed image. In the case of diffusion models, the

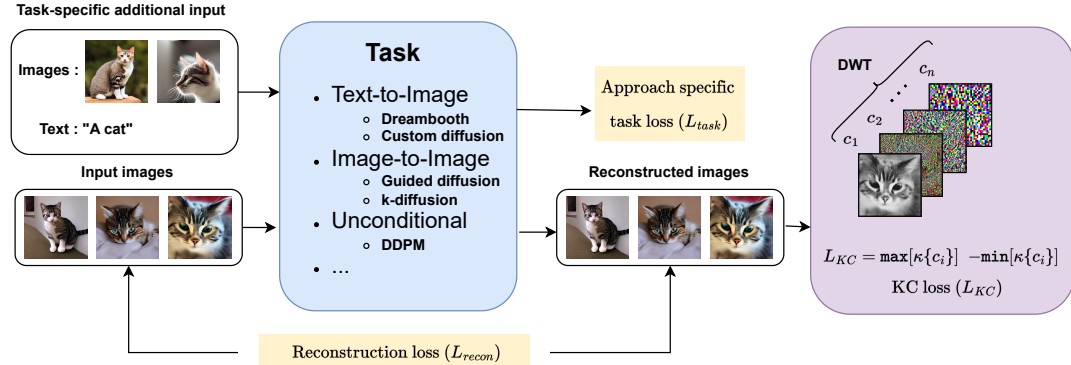

Figure 4: Overview of DiffNat. The proposed kurtosis concentration loss can be integrated to any diffusion based approach for various tasks (e.g., text-to-image generation (DreamBooth, Custom diffusion), super-resolution image-to-image generation (Guided diffusion, k-diffusion), unconditional image generation (DDPM)). In addition to the task specific losses, and general reconstruction loss, we incorporate the kurtosis concentration loss ($L_{KC}$), which operates on the reconstructed images and minimize the kurtosis deviation (i.e., $\max[\kappa[\{c_i\}] - \min[\kappa[\{c_i\}]]$) across Discrete Wavelet Transform (DWT) filtered version of the reconstructed image, Here, $c_1$, $c_2$ .. are DWT filtered version of the reconstructed image and $\kappa(x)$ denote kurtosis of $x$.

underlying denoising UNet is trained using mean squared error objective w.r.t the reconstructed image and the clean image. During inference, the reconstructed image is iteratively denoised and refined for $T$ steps to generate the final image with higher quality. Therefore, adding an objective to minimize the projection kurtosis of the reconstructed image, i.e., increasing the SNR (Lemma 2) would effectively lead to better denoising at each step and the final image would be of improved quality (shown in Fig. 3).

## 3.2 KURTOSIS CONCENTRATION (KC) LOSS

In this work, we leverage this property to introduce a novel loss function, viz., Kurtosis Concentration loss (KC loss) for training deep generative models. Unlike prior approaches Zhang & Lyu (2014), where the KC property has been used for noise estimation, source separation, etc., we utilize this property of natural images as a prior to train generative models for generating images with better perceptual quality. To validate our loss, we experiment with state-of-the-art generative models, i.e., diffusion models. Note that, our proposed loss can be integrated as a plug-and-play with any diffusion pipeline. We describe the basic diffusion pipeline and KC loss as follows.

Suppose, we need to train or finetune a diffusion model $f_\theta$ from input training images ($\{x\}$) with or without a conditioning vector $c$. The conditioning vector could be text, image, or none (in case of the unconditional diffusion model). The generated images obtained from $f_\theta$ given an initial noise map $\epsilon \sim N(0, I)$, a conditioning vector $c$ is given by $x_{gen} = f_\theta(x, \epsilon, c)$. Typically, the diffusion model is trained to minimize the $l2$ distance between the ground truth image ($x$) and the noisy image ($x_{gen}$) Dhariwal & Nichol (2021) or their corresponding latent in case of Latent Diffusion Model (LDM) Rombach et al. (2022). Without loss of generality, we are referring that as reconstruction loss ($L_{recon}$) between the ground-truth image ($x$) and the generated image ($x_{gen}$), denoted by,

$$L_{recon} = \mathbb{E}_{x,c,\epsilon}[\, ||x_{gen} - x||_2^2] \tag{5}$$

Note that for LDM, this will be the $l2$ distance between the corresponding latents. Now, we will describe the KC loss. Note, that the KC property holds across different bandpass transformed domains (DCT, DWT, fastICA) and we choose DWT because it is widely used due to its hierarchical structure and energy compaction properties E Woods & C Gonzalez (2008). Typically, DWT transforms images into LL (low-low), LH (low-high), HL (high-low), HH (high-high) frequency bands and each of the sub-bands contains several sparse details of the image. E.g., LL and HH subband contains a low-pass and high-pass filtered version of the image respectively Zhang & Lyu (2014) as shown in Fig. 5. The generated image $x_{gen}$ is then transformed using Discrete Wavelet Transform (DWT) with kernels $k_1, k_2, .., k_n$ producing filtered images $g_{gen,1}, g_{gen,2}, .., g_{gen,n}$ respectively, such that, $g_{gen,i} = F_{k_i}(x_{gen})$. Here, $F_l$ denotes the discrete wavelet transform with kernel $l$.

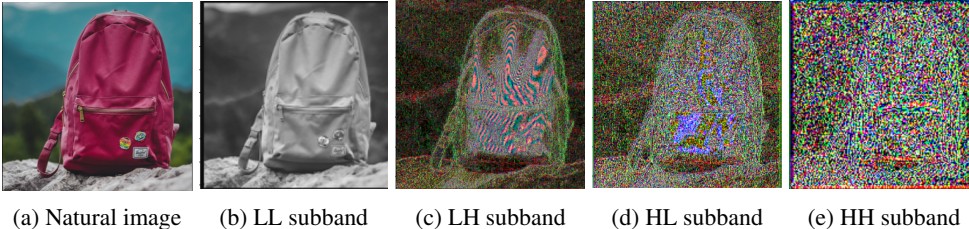

(a) Natural image     (b) LL subband     (c) LH subband     (d) HL subband     (e) HH subband

Figure 5: Wavelet transformed components of a natural image. LL and HH subband capture the low-frequency and high frequency details.

Now, kurtosis values of these $g_{gen,i}$ should be constant by the kurtosis concentration property, therefore, we minimize the difference between the maximum and minimum values of the kurtosis of $g_{gen,i}$'s to finetune the model using the loss,

$$L_{KC} = \mathbb{E}_{x,c,\epsilon}[(\max(\kappa(\{g_{gen,i}\})) - \min(\kappa(\{g_{gen,i}\})))] \tag{6}$$

Here, $\kappa(x)$ is kurtosis of $x$. Note that, this loss is quite generic and can be applied to both image or latent diffusion models for training. In the case of latent diffusion models, we need to transform the latent to image space, before applying this loss. In case of applying this loss to any task $T$ (DreamBooth, super-resolution, unconditional image generation), the overall loss ($L$) function would be, $L = L_{task} + L_{recon} + L_{KC}$, where $L_{task}$ is the task-specific loss.

## 4 EXPERIMENTS

We evaluate the efficacy of the proposed loss for three tasks - (1) personalized few-shot finetuning of diffusion model using text guidance, (2) unconditional image generation, and (3) image super-resolution.

### 4.1 TASK 1: PERSONALIZED FEW-SHOT FINETUNING USING TEXT GUIDANCE

In this section, we address the problem of finetuning the text-to-image diffusion model from a few examples for text-guided image generation in a subject-driven manner. Specifically, given only a few images (e.g., 3-5) of a particular subject without any textual description, our task is to learn the subject-specific details and generate new images of that particular subject in different conditions specified by the text prompt. Suppose we are given four samples of a dog backpack. Now the task is to finetune the text-to-image diffusion model given these four samples of the particular dog backpack such that it learns the concept/subject etc. During inference, the model has to generate images containing that particular dog backpack according to the text prompt.

To evaluate the efficacy of KC loss in this task, we build upon two popular methods, (1) Dreambooth Ruiz et al. (2022), and (2) Custom diffusion Kumari et al. (2022). In particular, we add KC loss to these frameworks while finetuning the denoising UNet to check whether the image quality improves and demonstrate the quality of generated images improves.

**Dataset and Metric.** We follow the dataset and experimental setup used by DreamBooth Ruiz et al. (2022). To evaluate the generated image quality with respect to the input image and the text prompt, we also use the subject fidelity metrics proposed by DreamBooth - (1) DINO, (2) CLIP-I, and (3) CLIP-T. We have also compared with another naturalness loss, i.e., LPIPS loss Zhang et al. (2018) as a baseline.

DreamBooth Ruiz et al. (2022) finetunes the stable diffusion model using the standard reconstruction loss and a prior preservation loss. However, it's prone to overfitting and some unnatural artifacts can be observed as shown in Fig. 1. For faster and lightweight training, custom diffusion Kumari et al. (2022) finetunes only the cross-attention module of the text-to-image stable diffusion model. We evaluate both approaches with/without KC loss on the same dataset for a fair comparison. When adding the proposed KC loss to these approaches, we obtain performance improvements in visual quality, i.e., FID Lucic et al. (2018), MUSIQ score Ke et al. (2021) as shown in Tab. 1. The qualitative results are shown in Fig. 6. We follow the same setup for the dreambooth and custom diffusion

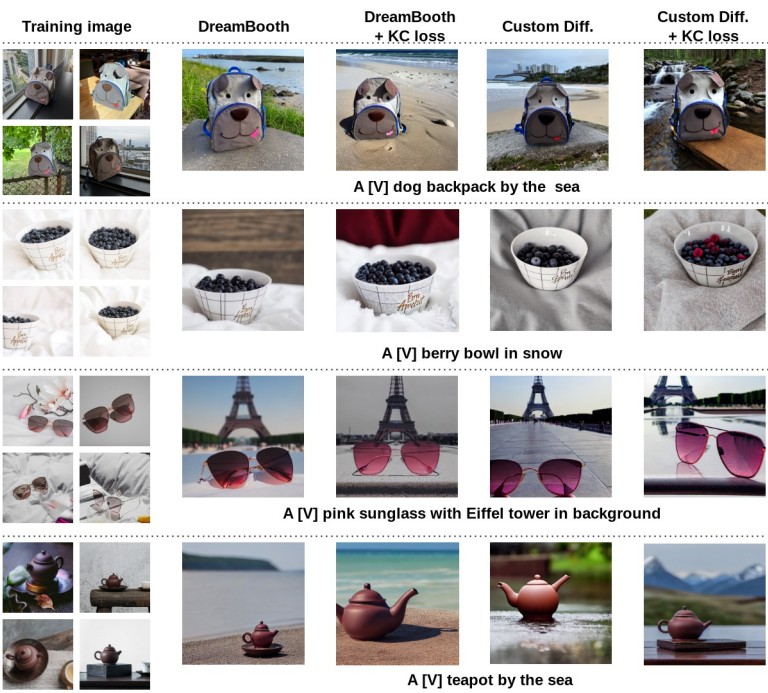

Figure 6: Comparison of DreamBooth, Custom diffusion with/without KC loss. Adding KC loss improves image quality for both DreamBooth and Custom diffusion, in terms of color vividness, contrast, and lighting consistency.

baselines. Additionally, for KC loss, we decompose the reconstructed images using 27 'Daubechies' filter banks, and get the average deviation of the kurtosis values as a loss function. More training details will be provided in the supplementary material.

Table 1: Comparison of Personalized few-shot finetuning task

| Method | Image quality | | Subject fidelity | | Prompt fidelity |
|---|---|---|---|---|---|
| | FID score ↓ | MUSIQ score ↑ | DINO ↑ | CLIP-I ↑ | CLIP-T ↑ |
| DreamBooth Ruiz et al. (2022) | 111.76 | 68.31 | 0.65 | 0.81 | 0.31 |
| DreamBooth Ruiz et al. (2022) + LPIPS | 108.23 | 68.39 | 0.65 | 0.80 | 0.32 |
| DreamBooth + KC loss(Ours) | **100.08** | **69.78** | **0.68** | **0.84** | **0.34** |
| Custom Diff. Kumari et al. (2022) | 84.65 | 70.15 | 0.71 | 0.87 | 0.38 |
| Custom Diff. Kumari et al. (2022) + LPIPS | 80.12 | 70.56 | 0.71 | 0.87 | 0.37 |
| Custom Diff. + KC loss(Ours) | **75.68** | **72.22** | **0.73** | **0.88** | **0.40** |

**Human evaluation.** Since the perceptual quality is quite subjective, automatic metrics do not correlate well with the perceptual studies Zhang et al. (2018). To verify that the improved scores actually correspond to better quality images, we evaluate our approach using human preference study through Amazon Mechanical Turk. Specifically, we performed two human evaluation tasks - (1) subject fidelity assessment and (2) image quality ranking.

In the subject fidelity assessment, we conduct Two Alternative Forced choice (2AFC) experiment setup. In particular, we show a pair of images containing the real image and the edited image using KC loss and asked the user the question : "How similar are these two objects?", with 10 options ranging from "extremely likely" to "extremely unlikely"( with "0" being "extremely unlikely" to 10 being "extremely likely"). We test this with 423 samples with 10 human evaluations per sample, total-ing 4230 tasks. We show the aggregate response in Fig. 7, which reveals that adding our proposed loss retains subject fidelity in most cases.

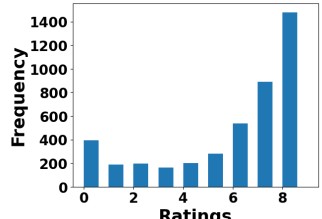

Figure 7: Subject fidelity assessment by user study. The ratings ranges from "0" being "extremely unlikely" to 10 being "extremely likely". We observe from the plot that most of the users find DiffNat preserves subject fidelity. The average rating is 5.8, which is "moderately likely" to "highly likely".

Next, we provide 30 examples of natural images and corresponding generated images using Dreambooth, Custom diffusion, and our method, and asked the question : "which of the edited images is of best visual quality considering factors including image quality and preserving the identity of the original image?" We evaluate this by 50 users, totaling 1500 questionnaires and the aggregate response reveals that DiffNat-generated images outperform the baselines by a large margin (i.e., 50.4%,) where the available options are { 'DiffNat', 'Dreambooth', 'Custom diffusion', 'None is satisfactory'}, which shows that our approach outperforms the baseline approaches.

## 4.2 TASK 2: UNCONDITIONAL IMAGE GENERATION

Unconditional image generation does not require any text or image guidance. It simply tries to learn the training data distribution through a generative model (here we focus on the diffusion model) and generates samples similar to the training data distribution. Denoised Diffusion Probabilistic Model (DDPM) Ho et al. (2020) is a parameterized Markov chain that is trained to generate matched data distribution through variational inference. The denoising takes place in the image space and produces better image quality compared to GANs.

We incorporate our proposed KC loss in this framework and obtain even better perceptual quality on various diverse datasets in terms of FID and MUSIQ score. The experimental results are shown in Tab. 2 and Fig. 8. Note that, in this approach, we integrate the KC loss directly into image space, which shows the flexibility of our proposed loss. We have experimented with Oxford-flowers Nilsback & Zisserman (2006), celebAfaces Zhang et al. (2020) and CelebAHQ Karras et al. (2017) datasets and obtained consistent improvements on image quality as shown in Tab. 2. Qualitative analysis in Fig. 8 verify that integrating KC loss improves image quality in terms of details, contrast, and color vividness.

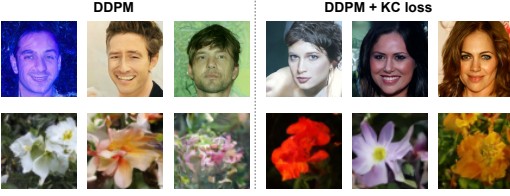

Figure 8: Comparison of unconditional image generation (DDPM) with/without KC loss. Integrating KC loss significantly improve image quality, whereas DDPM generated images have unnatural image artifacts.

Table 2: Comparison of unconditional image generation task

| Method | Oxford flowers | | Celeb-faces | | CelebAHQ | |
|---|---|---|---|---|---|---|
| | FID score ↓ | MUSIQ score ↑ | FID score ↓ | MUSIQ score ↑ | FID score ↓ | MUSIQ score ↑ |
| DDPM Ho et al. (2020) | 243.43 | 20.67 | 202.67 | 19.07 | 199.77 | 46.05 |
| DDPM Ho et al. (2020) + LPIPS | 242.62 | 20.80 | 201.55 | 19.21 | 197.17 | 46.15 |
| DDPM + KC loss(Ours) | **237.73** | **21.13** | **198.23** | **19.52** | **190.59** | **46.83** |

## 4.3 TASK 3: IMAGE SUPER-RESOLUTION

Image super-resolution typically takes the form of a conditional generation task, leveraging a low-resolution image as an additional condition for the diffusion model. In this study, we establish two state-of-the-art diffusion pipelines as baselines for comparison. Guided diffusion (GD) Dhariwal & Nichol (2021) directly takes the low-resolution image as a condition and performs the diffusion operation in the pixel space. Additionally, we also explore the latent diffusion model (LDM) Rombach et al. (2022) that operates in the latent space of a pre-trained VQ-VAE Esser et al. (2021). We introduce conditioning by utilizing the latent embedding of the low-resolution image with this model, referring to it as conditional-LDM (cLDM).

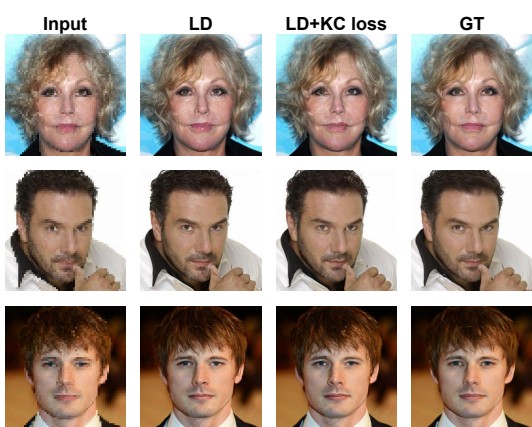

Figure 9: Image super-resolution quality improves adding KC loss to k-diffusion (LD) framework. Generated images show better quality in terms of overall image smoothness (first row), finer details like eyes (second and third row).

Note that, as GD operates in the pixel space, we directly add the proposed KC loss to the output of the denoising UNet. Conversely, for cLDM, we initially convert the latent embedding to image space using the pre-trained decoder and integrate the KC loss on the output of the decoder. For training, we use the standard FFHQ dataset Karras et al. (2017), which contains 70k high-quality images. Specifically, we address the task of $\times 4$ super-resolution where the GT images are of resolution $256 \times 256$. We evaluate randomly sampled 1000 images from CelebA-Test dataset Karras et al. (2017) under the same $\times 4$-SR setting.

Table 3: Comparison of image super-resolution task

| Method | Image quality | | | | |
|---|---|---|---|---|---|
| | FID score ↓ | PSNR ↑ | SSIM ↑ | LPIPS ↓ | MUSIQ score ↑ |
| GD Dhariwal & Nichol (2021) | 121.23 | 18.13 | 0.54 | 0.28 | 57.31 |
| GD Dhariwal & Nichol (2021) + LPIPS | 119.81 | 18.22 | 0.54 | 0.27 | 57.42 |
| GD + KC loss(Ours) | **103.19** | **18.92** | **0.55** | **0.26** | **58.69** |
| LD. Karras et al. (2022) | 95.83 | 19.16 | 0.56 | 0.26 | 59.57 |
| LD. Karras et al. (2022) + LPIPS | 92.77 | 19.42 | 0.57 | 0.25 | 59.82 |
| LD + KC loss(Ours) | **83.34** | **20.25** | **0.58** | **0.22** | **61.20** |

Since, the proposed KC loss improves image quality, it is inherently applicable for this task. We integrate KC loss in the SOTA super-resolution diffusion models Dhariwal & Nichol (2021); Rombach et al. (2022), and obtained performance improvement in perceptual quality as shown in Tab. 3. In the qualitative results shown in Fig. 10 and Fig. 9, we observe that adding KC loss improves the image quality and finer details, e.g., eye structure, texture, lighting etc.

## 4.4 COMPARISON OF REAL VS SYNTHETIC DETECTION

To perform the robustness analysis of the proposed loss, we also perform the following experiment. We train a classifier (2-layer MLP on top of pre-trained ResNet feature extractor) to distinguish real vs synthetic, where 'real' comes from natural image belongs to the DreamBooth dataset and 'synthetic' comes from diffusion model generated images from algorithm X or X + KC loss. Here 'X' can be 'DreamBooth' or 'Custom diffusin". For testing, we select non-overlapping test samples for both natural and diffusion generated images. When tested on DreamBooth and Custom diffusion, we observe that adding KC loss decrease the real vs synthetic classification accuracy as shown in Tab. 4. This indicates that generated images are of superior perceptual quality, exhibiting a greater degree of "naturalness" to both human observers and machine algorithms alike.

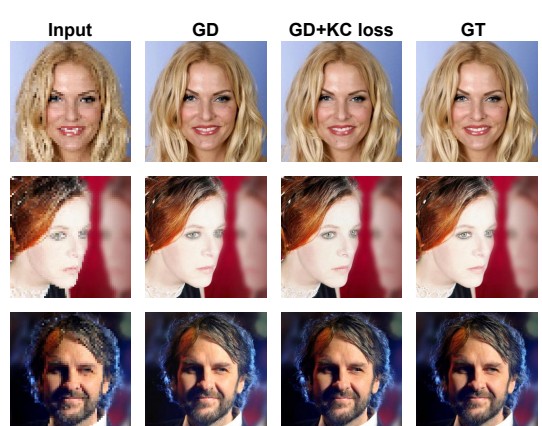

Figure 10: Image super-resolution quality improves adding KC loss to the guided diffusion (GD) framework. Generated images show better quality in terms of overall image details (first row), finer details like eyes (second row), and overall color and brightness (third row).

## 5 CONCLUSION

Although diffusion models have significantly advanced in creating naturalistic images, these images can have unnatural artifacts, especially in the cases of few-shot finetuning of large-scale text-to-image diffusion models. We leverage the kurtosis concentration property of natural images to define a novel and generic loss function

Table 4: Comparison of real vs synthetic detection

| Method | Accuracy |
|---|---|
| DreamBooth Ruiz et al. (2022) | 93.33% |
| DreamBooth Ruiz et al. (2022) + KC loss | 66.66% |
| Custom Diffusion Kumari et al. (2022) | 94.16 % |
| Custom Diffusion Kumari et al. (2022) + KC loss | 92.5% |

in order to preserve the "naturalness" of generated images. Kurtosis concentration property suggests that the kurtosis values across different bandpass versions of the natural image tend to be constant. The proposed kurtosis concentration loss minimizes the gap between the maximum and minimum value of the kurtosis across different DWT filtered versions of the image. We show this loss improves image quality for diverse generative tasks - (1) personalized few-shot finetuning of text-to-image diffusion model, (2) unconditional image generation, and (3) image super-resolution. We also conduct human studies to validate our approach.

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

# A  APPENDIX

In this supplementary material, we will provide the following details.

1. Training details.
2. Theoretical justification.
3. Additional experimental results.
4. Failure cases.
5. Computation complexity
6. Training time analysis
7. Kurtosis analysis
8. Convergence analysis
9. Qualitative analyis
10. Experiments on image super-resolution

## B  TRAINING DETAILS

The training details of finetuning the diffusion model for various tasks have been provided here. For personalized few-shot finetuning, we consider two methods - Dreambooth Ruiz et al. (2022) and Custom diffusion Kumari et al. (2022). For fair comparison, we applied both the approaches on the dataset and setting introduced by Dreambooth. The dataset contains 30 subjects (e.g., backpack, stuffed animal, dogs, cats, sunglasses, cartoons etc) and 25 prompts including 20 re-contextualization prompts and 5 property modification prompts. DINO, which is the average pairwise cosine similarity between the ViT-S/16 DINO embeddings Caron et al. (2021) of the generated and real images. (2) CLIP-I, i.e., the average pairwise cosine similarity between CLIP Radford et al. (2015) embeddings of the generated and real images. To measure the prompt fidelity, we use CLIP-T, which is the average cosine similarity between prompt and image CLIP embeddings.

For unconditional image generation, we have experimented on oxford flowers, CelebAfaces and CelebAHQ datasets. Image quality has been measured by FID and MUSIQ score.

In case of image super-resolution, we experimented with guided diffusion Dhariwal & Nichol (2021) and latent diffusion Karras et al. (2022) pipelines. We use FFHQ dataset for training, and test on a subset of 1000 images from CelebAHQ test set for x4 super-resolution task. The hyperparameter details are given in Tab. 5.

Table 5: Hyperparameters

| Hyperparameter | Values |
| --- | --- |
| Coefficient of $L_{recon}$ | 1 |
| Coefficient of $L_{prior}$ | 1 |
| Coefficient of $L_{KC}$ | 1 |
| Learning rate | $10^{-5}$ |
| Batch size (Dreambooth, Custom diffusion) | 8 |
| Batch size (DDPM) | 125 |
| Batch size (GD) | 16 |
| Batch size (LD) | 9 |
| Text-to-image diffusion model | Stable Diffusion-v1 Rombach et al. (2022) |
| Number of class prior images (Dreambooth, Custom diffusion ) | 10 |
| Number of DWT components | 25 |

## C  THEORETICAL JUSTIFICATION

Here we provide theoretical analysis of the Lemmas mentioned in the main paper.

**Lemma 3** *A Gaussian scale mixture (GSM) vector $x$ with zero mean has the following probability density function:*

$$p(x) = \int_0^\infty \mathcal{N}(x; 0, z\Sigma_x) p_z(z) dz \tag{7}$$

*and its projection kurtosis is __constant__ with respect to the projection direction w, i.e.,*

$$\kappa(w^T x) = \frac{3 var_z\{z\}}{\mathcal{E}_z\{z\}^2} \tag{8}$$

*where $\mathcal{E}_z\{z\}$ and $var_z\{z\}$ are the mean and variance of latent variable $z$ respectively.*

*Proof.* Marginal distribution of the projection of $x$ on non-zero vector $w$ is given by,

$$p_w(t) = \int_{x:w^T x=t} p(x) dx$$

$$= \int_z p_z(z) dz . \int_{x:w^T x=t} \frac{1}{\sqrt{(2\pi z)^d |det(\Sigma_x)|}} exp(-\frac{x^T \Sigma_x^{-1} x}{2z}) dx$$

$$= \int_z \mathcal{N}_t(0, z w^T \Sigma_x w) p_z(z) dz$$

Note that, the last equality holds from the marginalization property of Gaussian, i.e., $X \approx \mathcal{N}(\mu, \Sigma)$, then, $AX \approx \mathcal{N}(A\mu, A\Sigma A^T)$.

The variance of $w^T x$,

$$\mathcal{E}_t\{t^2\} = \int_z p_z dz \int_t t^2 \mathcal{N}_t(0, zw^T\Sigma_x w)dz$$
$$= w^T\Sigma_x w \int_z z p_z dz$$
$$= w^T\Sigma_x w \mathcal{E}_z\{z\}$$

The fourth order moment of $w^T x$,

$$\mathcal{E}_t\{t^4\} = \int_z p_z dz \int_t t^4 \mathcal{N}_t(0, zw^T\Sigma_x w)dz$$
$$= 3(w^T\Sigma_x w)^2 \int_z z^2 p_z dz$$
$$= 3(w^T\Sigma_x w)^2 \mathcal{E}_z\{z^2\}$$

We utilize the property that $\mathcal{N}_t(0, \sigma^2)$ has a fourth order moment of $3\sigma^4$.

Finally, the kurtosis becomes,

$$\kappa(w^T x) = \frac{\mathcal{E}_t\{t\}^4}{\mathcal{E}_t\{t\}^2} - 3$$
$$= \frac{3\mathcal{E}_z\{z\}^2}{\mathcal{E}_z\{z\}^2} - 3$$
$$= \frac{3(\mathcal{E}_z\{z^2\} - \mathcal{E}_z\{z\}^2)}{\mathcal{E}_z\{z\}^2}$$
$$= \frac{3var_z\{z\}}{\mathcal{E}_z\{z\}^2}$$

**Lemma 4** *If the noisy version of the natural image is denoted by, y = x + n, where x is a whitened GSM vector (normalized natural image) and n is a zero-mean white Gaussian noise with variance $\sigma^2 I$, x and n are mutually independent of each other, then the projection kurtosis of y, $\kappa(w^T y)$ can be expressed as:*

$$\kappa(w^T y) = \kappa(w^T x)\left(1 - \frac{c}{SNR(y)}\right)^2 = \frac{3var_z\{z\}}{\mathcal{E}_z\{z\}^2}\left(1 - \frac{c}{SNR(y)}\right)^2 \quad (9)$$

*where Signal-to-Noise Ratio (SNR) is defined as, $SNR(y) = \frac{\sigma^2(y)}{\sigma^2(n)}$ and c is a constant.*

*Proof.* Here, we provide the proof of Lemma 1, mentioned in the main paper. Without loss of generality, we strat by assuming, $\mathcal{E}_x x = 0$, since the mean can be easily subtracted from the data. We also assume that n is a zero-mean white Gaussian noise with variance $\sigma^2 I$, x and n are mutually independent of each other.

$$\sigma^2(w^T n) = w^T \mathcal{E}_z\{zz^T\}w = \sigma^2 w^T w = \sigma^2$$
$$\sigma^2(w^T x) = w^T \mathcal{E}_x\{xx^T\}w = w^T\Sigma_x w$$
$$\sigma^2(w^T y) = \sigma^2(w^x y) + \sigma^2(w^T n) = w^T\Sigma_x w + \sigma^2$$

Since $n$ is a white Gaussian, $x$ and $n$ are independent, then $w^T x$ and $w^T n$ Therefore,

$$\sigma^2(w^T y) = \sigma^2(w^T x) + \sigma^2(w^T n) \quad (10)$$

.

Similarly, for fourth order moment, using the additivity of cumulants of independent variables (since $x$ and $n$ are independent) Papoulis & Unnikrishna Pillai (2002), we obtain,

$$
\begin{aligned}
\kappa(w^T y)(\sigma^2(w^T y))^2 &= \kappa(w^T x)(\sigma^2(w^T x))^2 + \kappa(w^T n)(\sigma^2(w^T n))^2 \\
&= \kappa(w^T x)(\sigma^2(w^T x))^2
\end{aligned}
\tag{11}
$$

Since, For Gaussian, $\kappa(n) = 0$

By rearranging, we have,

$$
\begin{aligned}
\kappa(w^T y) &= \kappa(w^T x).\big(\frac{\sigma^2(w^T x)}{\sigma^2(w^T y)}\big)^2 \\
&= \kappa(w^T x).\big(\frac{\sigma^2(w^T y) - \sigma^2}{\sigma^2(w^T y)}\big)^2 \\
&= \kappa(w^T x).\big(1 - c.\frac{\sigma^2}{\sigma^2(y)}\big)^2 \\
&= \frac{3 var_z\{z\}}{\mathcal{E}_z\{z\}^2}.\big(1 - \frac{c}{SNR(y)}\big)^2
\end{aligned}
$$

Here, Signal-to-Noise Ratio (SNR) is defined as, $\text{SNR}(y) = \frac{\sigma^2(y)}{\sigma^2(n)}$.

## D    ADDITIONAL EXPERIMENTAL RESULTS

In Fig. 11, we visualize some of the DiffNat generated images using various text-prompts. The generated images capture the context of the text-prompt and also retain naturalness. We have also provided qualitative comparison w.r.t Dreambooth in Fig. 12.

## E    FAILURE CASES

We also present some of the failure cases of DiffNat in Fig. 13. E.g., our model fails to generate images of "A [V] berry bowl with the Eiffel Tower in the background", but actually generates images with "the Eiffel Tower" in the berry bowl. Similarly, the model fails to generate "A cube shaped [V] can", since these object do not appear in the training set. The model also fails to generate "A [V] cat on top of a purple rug in a forest" and instead generated some version of purple cat.

## F    COMPUTATION COMPLEXITY

Here we analyze the computational complexity of the proposed KC loss. Suppose, given a batch of N images. We need to perform DWT of each images using k different filters. Since, DWT for 'haar' wavelet can be done in linear time, the complexity of performing DWT with k filters can be done in $\mathcal{O}(Nk)$ time. Now, calculating the difference between maximum and minimum kurtosis can be done in linear time, therefore, the computational complexity of calculating KC loss is $\mathcal{O}(Nk)$. This minimal overhead of computing KC loss can be observed in the training time analysis provided next.

## G    TRAINING TIME ANALYSIS

The run time analysis has been provided in Table. 6. Note that the experiments for Dreambooth, Custom diffusion, DDPM have been performed on a single A5000 machine with 24GB GPU. We

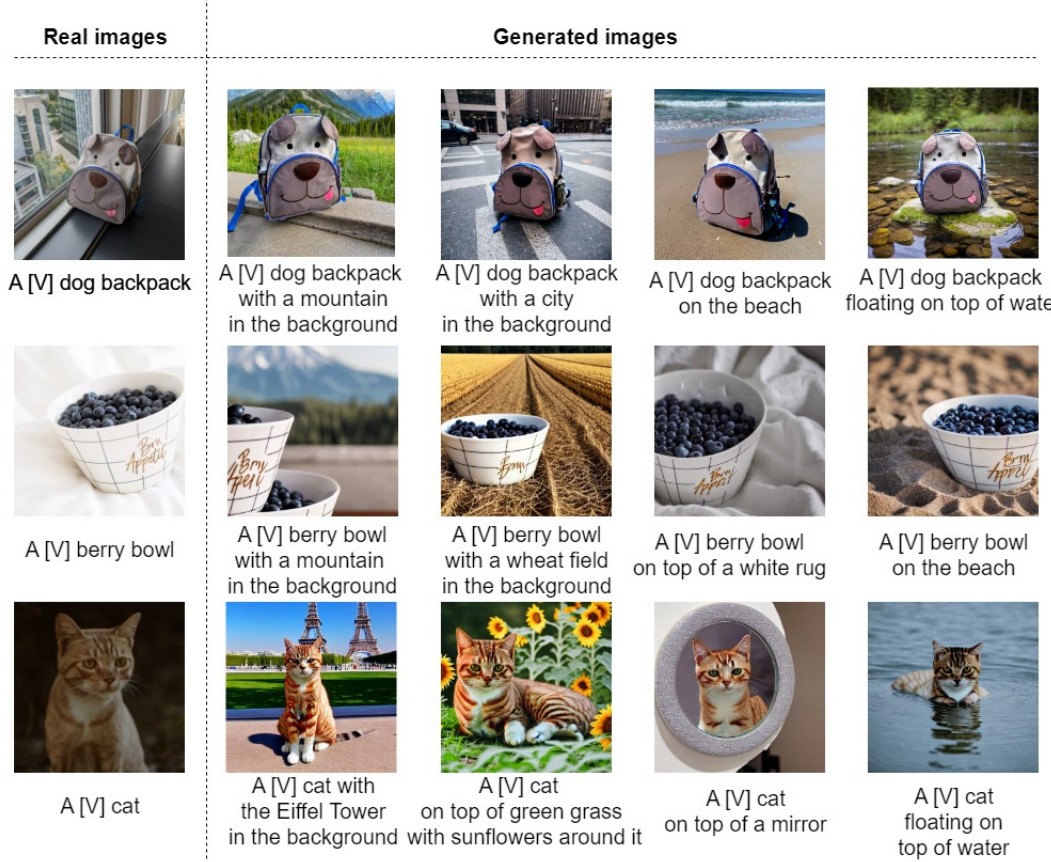

Figure 11: DiffNat generated images. The task is to learn a unique identifier ("A [V] dog backpack") of the training images and generate variations w.r.t. background, lighting conditions etc. The generated images look natural in different background context, e.g., "A [V] dog backpack on the beach/ with a city in the background etc". The generated images are of high quality.

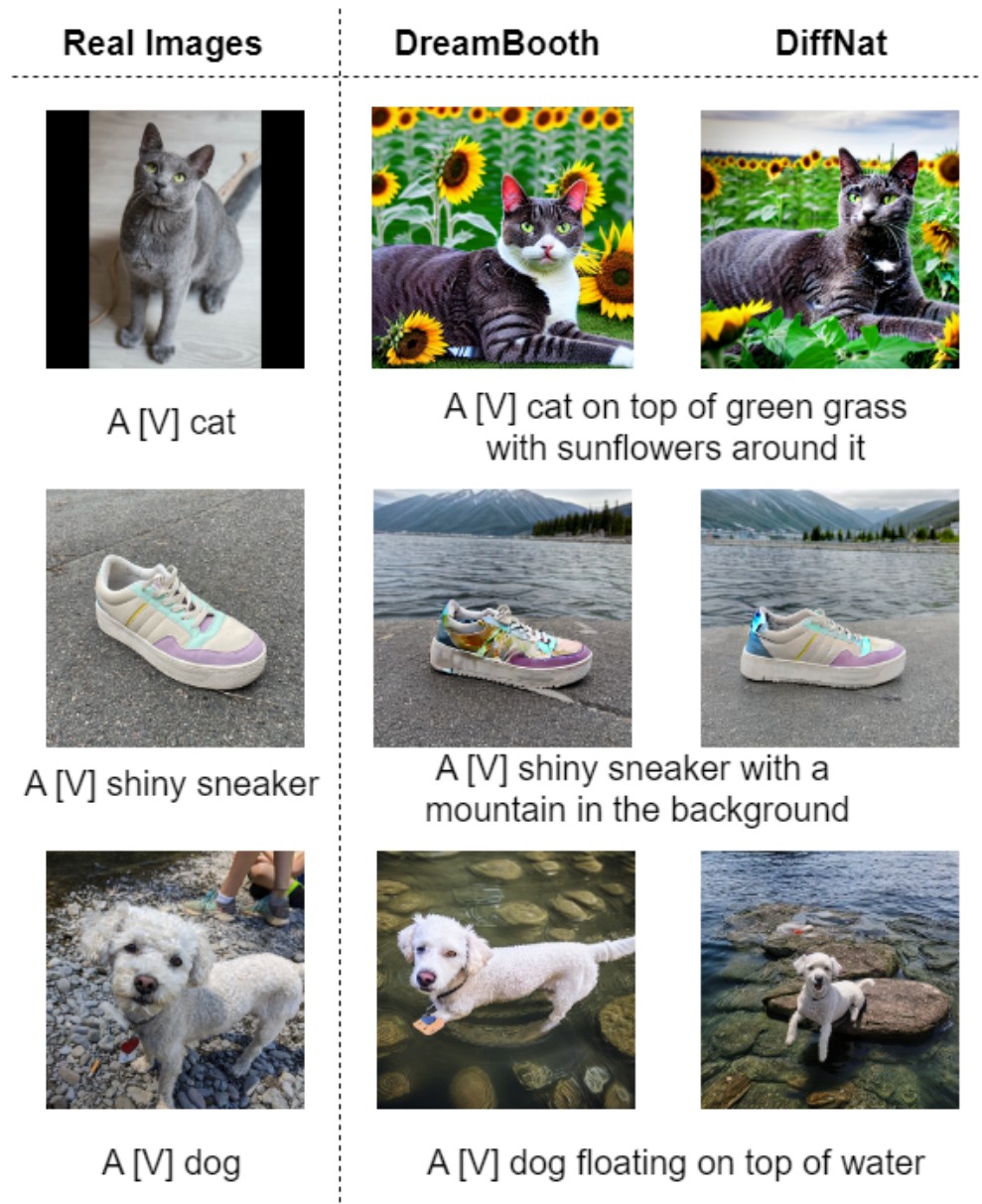

Figure 12: Comparison of DreamBooth and DiffNat. DiffNat generated images have better visual quality.

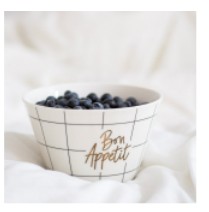 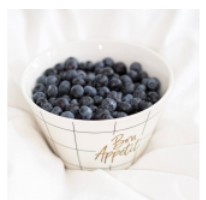 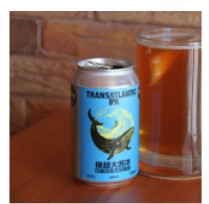 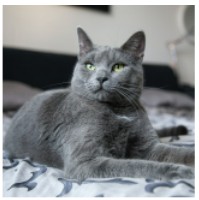

Real images

Generated images

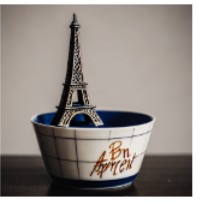 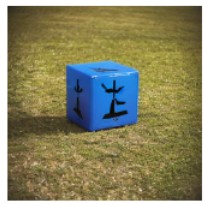 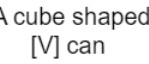 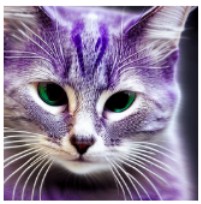

A [V] berry bowl with the Eiffel Tower in the background

A [V] berry bowl on top of green grass with sunflowers around it

A cube shaped [V] can

A [V] cat on top of a purple rug in a forest

Figure 13: Failure cases of DiffNat. Instead of generating "A [V] berry bowl with the Eiffel Tower in the background", our method generates image with the Eiffel Tower in the berry bowl. Also, while generating "A [V] cat on top of a purple rug in a forest", it generates a purple [V] cat, which shows the color bias w.r.t the text-prompt of the model.

have performed guided diffusion (GD) and latent diffusion (LD) experiments on a server of 8 24GB A5000 GPUs. The experimental results in Table. 6 show that incorporating KC loss induces minimum training overhead.

## H  KURTOSIS ANALYSIS

To verify the efficacy of the proposed KC loss, we perform average kurtosis analysis in this section. we compute the average kurtosis deviation of DWT filtered version of images from the dataset and plot them in Fig. 15, Fig. 16 and Fig. 17. E.g., in case of dreambooth task, we compute the kurtosis statistics of bandpass filtered version of natural images from Dreambooth dataset, images generated by Dreambooth and images generated by DiffNat (i.e., adding KC loss) and plot it in Fig. 15. We observe that the Dreambooth generated images (Fig. 15 (a)) have highest kurtosis deviation. The average deviation is least for natural images (Fig. 15 (c)) and adding KC loss reduces the kurtosis deviation (Fig. 15 (b)). Similar trends can be observed for DDPM (Fig. 16), guided diffusion (Fig. 17) as well. Adding KC loss improves image quality has been verified both qualitatively and quantitatively in the paper. This analysis verifies minimizing kurtosis loss improves diffusion image quality.

## I  CONVERGENCE ANALYSIS

The main idea of the diffusion model is to train a UNet, which learns to denoise from a random noise to a specific image distribution. More denoising steps ensure a better denoised version of the image, e.g., DDPM Ho et al. (2020), LDM Karras et al. (2022). In proposition 1 (main paper), we show that minimizing projection kurtosis further denoise input signals. Therefore, KC loss helps in the denoising process and improves the convergence speed. We have shown that adding KC loss improves the loss to converge faster for Dreambooth task in Fig. 14.

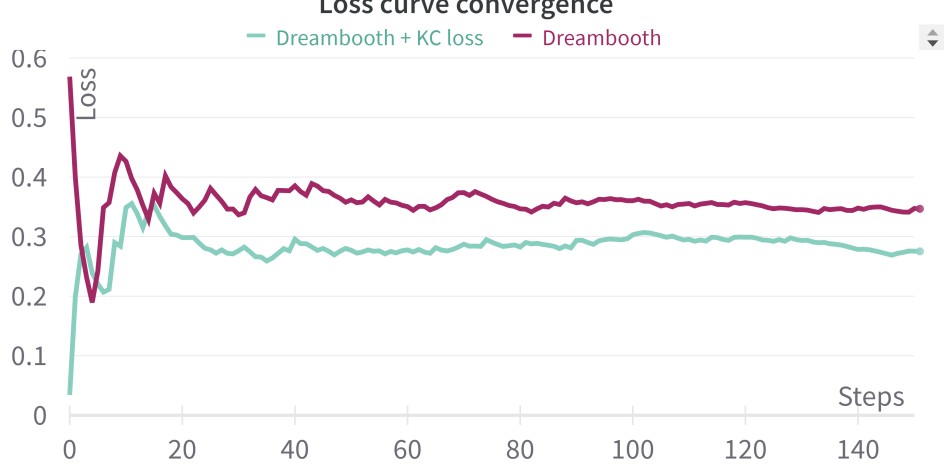

Figure 14: Loss curve convergence of Dreambooth.

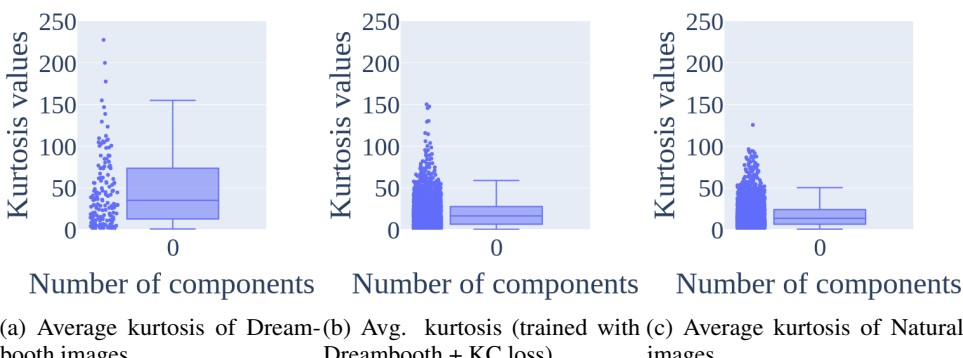

(a) Average kurtosis of Dream-  (b) Avg. kurtosis (trained with  (c) Average kurtosis of Natural
booth images                     Dreambooth + KC loss)             images

Figure 15: Average kurtosis analysis of Dreambooth, DiffNat and natural images over the dataset
used in Dreambooth. From this analysis, it is evident that Dreambooth generated images have higher
kurtosis deviation. Integrating KC loss reduces the kurtosis deviation to preserve the naturalness of
the generated images. Natural images have more concentrated kurtosis values.

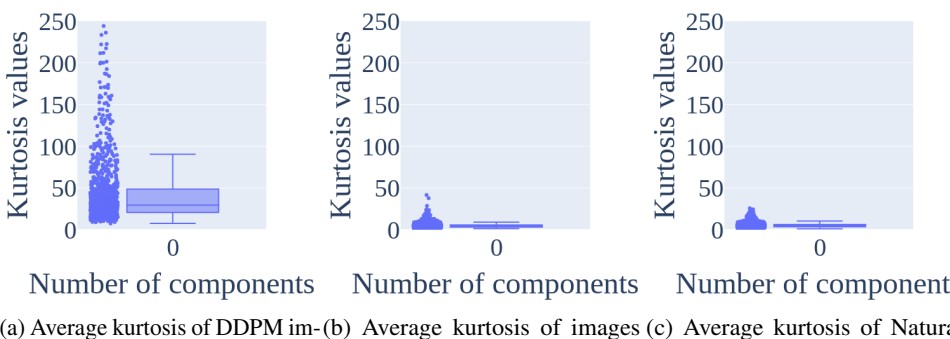

(a) Average kurtosis of DDPM im-  (b) Average kurtosis of images  (c) Average kurtosis of Natural
ages                              trained with DDPM + KC loss     images

Figure 16: Average kurtosis analysis of DDPM framework trained on Oxford flowers dataset. From
this analysis, it is evident that DDPM generated images have higher kurtosis deviation. Integrating
KC loss reduces the kurtosis deviation to preserve the naturalness of the generated images. Natural
images have more concentrated kurtosis values.

Table 6: Training time analysis

| Method | dataset | Training time |
| --- | --- | --- |
| DreamBooth Ruiz et al. (2022) | 5-shot finetuning | 10 min 21s |
| DreamBooth Ruiz et al. (2022) + KC loss | 5-shot finetuning | 11 min 30s |
| Custom Diffusion Kumari et al. (2022) | 5-shot finetuning | 6m 43s |
| Custom Diffusion Kumari et al. (2022) + KC loss | 5-shot finetuning | 7m 11s |
| DDPM Ho et al. (2020) | CelebAfaces | 2d 8h 21m |
| DDPM Ho et al. (2020) + KC loss | CelebAfaces | 2d 9h 19m |
| DDPM Ho et al. (2020) | CelebAHQ | 21h 48m |
| DDPM Ho et al. (2020) + KC loss | CelebAHQ | 22h 40m |
| DDPM Ho et al. (2020) | Oxford flowers | 6h 17m |
| DDPM Ho et al. (2020) + KC loss | Oxford flowers | 6h 39m |
| GD Dhariwal & Nichol (2021) | FFHQ | 23h 10m |
| GD Dhariwal & Nichol (2021) + KC loss | FFHQ | 1d 1h 29m |
| LD Karras et al. (2022) | FFHQ | 20h 15m |
| LD Karras et al. (2022) + KC loss | FFHQ | 22h 40m |

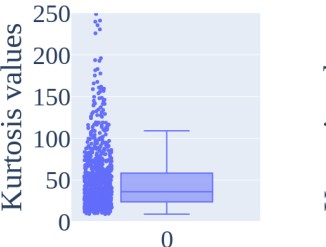 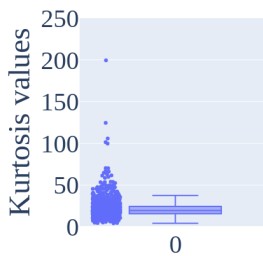 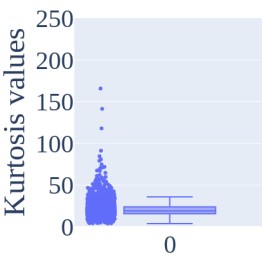

(a) Average kurtosis of GD gener-  (b) Average kurtosis of images  (c) Average kurtosis of Natural
ated images                trained with GD + KC loss      images

Figure 17: Average kurtosis analysis of guided diffusion (GD) framework trained on FFHQ dataset. From this analysis, it is evident that GD generated images have higher kurtosis deviation. Integrating KC loss reduces the kurtosis deviation to preserve the naturalness of the generated images. Natural images have more concentrated kurtosis values.

## J  QUALITATIVE ANALYSIS

In this section, we provide more qualitative analysis to show that adding KC loss improves image quality. Zoomed view of the generated images are shown to compare w.r.t the baselines in Fig. 18, Fig. 19, Fig. 20, Fig. 21, Fig. 22, Fig. 23, Fig. 24, Fig. 25. Details are provided in the caption.

## K  EXPERIMENTS ON IMAGE SUPER-RESOLUTION

In this section, we provide more experimental results for image super-resolution task. This includes quantitative results and human evaluation.

### K.1  QUANTITATIVE RESULTS

In addition to the super resolution task (x4) shown in the main paper, we conduct experiments for x2 and x8 tasks as well in the same setting. The ground-truth images are of size 256 X 256. Therefore, x2 task performs image super-resolution from 128 X 128 $\rightarrow$ 256 X 256 and x8 task performs image super-resolution from 32 X 32 $\rightarrow$ 256 X 256 and the corresponding experiments are shown in Table 7 and Table 8 respectively. For training, we use standard FFHQ dataset Karras et al. (2017), and evaluation is performed on CelebA-Test dataset Karras et al. (2017). We observe that adding KC loss improves image quality quantitatively both for guided diffusion (GD) and latent diffusion (LD). Qualitative results are shown in Fig. 22, Fig. 23, Fig. 24 and Fig. 25. Next, we also perform human study to validate our approach.

**Dreambooth**

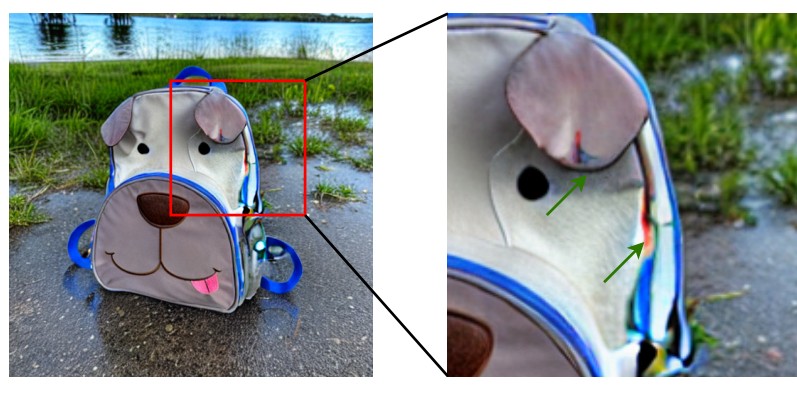

**Dreambooth + KC Loss**

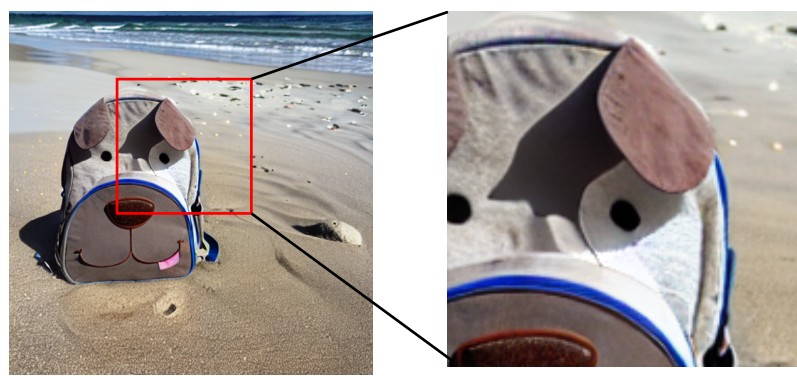

Figure 18: Qualitative comparison of with/without KC loss in Dreambooth. The bottom image (with KC loss) shows better image quality and shadows (best viewed in color).

Table 7: Comparison of image super-resolution (x2) task

| Method | Image quality | | | | |
|---|---|---|---|---|---|
| | FID score ↓ | PSNR ↑ | SSIM ↑ | LPIPS ↓ | MUSIQ score ↑ |
| GD Dhariwal & Nichol (2021) | 100.2 | 19.4 | 0.62 | 0.25 | 58.12 |
| GD + KC loss(Ours) | **80.9** | **20.2** | **0.66** | **0.20** | **59.91** |
| LD. Karras et al. (2022) | 82.45 | 21.2 | 0.64 | 0.24 | 60.23 |
| LD + KC loss(Ours) | **70.12** | **22.3** | **0.70** | **0.18** | **62.15** |

### K.2 HUMAN EVALUATION

We conduct human evaluation of image super-resolution task to compare guided diffusion (GD)/ latent diffusion (LD) and adding KC loss to the corresponding counterpart (DiffNat). We provide 20 examples of natural images and corresponding generated images using GD, LD and our method DiffNat (i.e., adding KC loss) and asked the following question to amazon mechanical turks: "which of the generated images is of best visual quality considering factors include image quality and preserving the identity of the original image?" Similar to Dreambooth task, we evaluate this by 50 users, totalling 1000 questionnaires. The available options are { 'DiffNat', 'GD/LD', 'None is satisfactory' }. The aggregate response shows that DiffNat generated images are of better image quality compared to the baselines, as shown in Fig. 26. Therefore, we verified the improved image quality quantitatively, qualitatively and through human evaluation as well. Note that, human evaluation is not applicable for unconditional image generation task since there is no one-to-one correspondence between the training images and the generated images. It will be ambiguous for the human observers to compare quality between approaches. Therefore, we abstain ourselves from performing human evaluation for this task. However, the quantitative and qualitative analysis exhibit the efficacy of our approach.

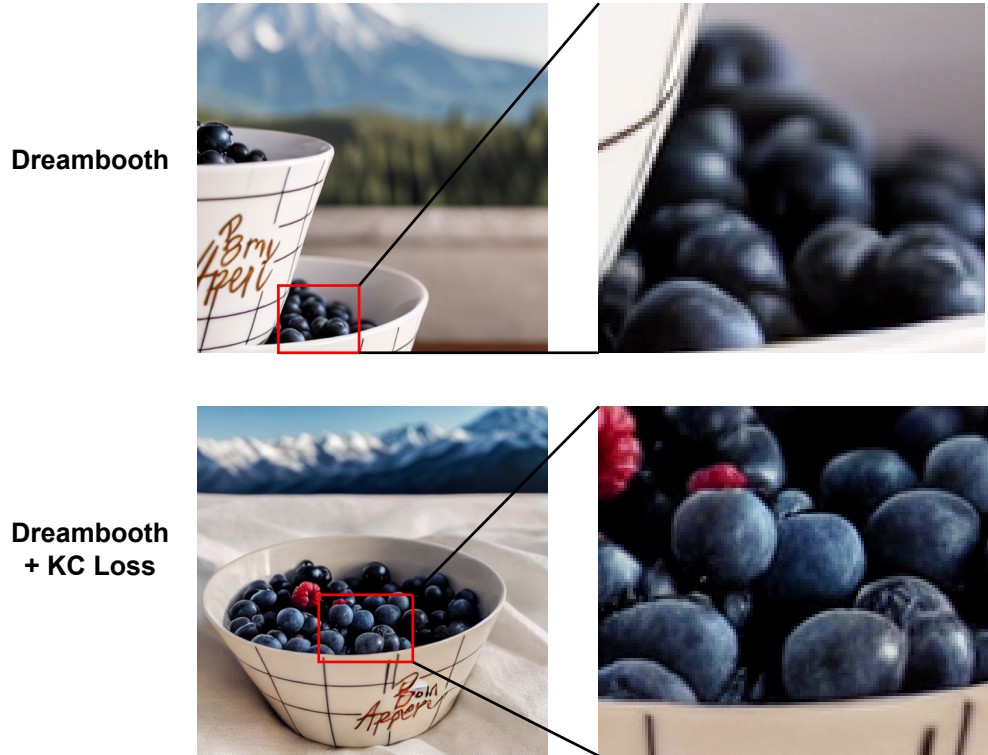

Figure 19: Qualitative comparison of with/without KC loss in Dreambooth. The bottom image (with KC loss) shows better image quality and reflections on the bowl full of berries (best viewed in color).

Table 8: Comparison of image super-resolution (x8) task

| Method | Image quality | | | | |
|---|---|---|---|---|---|
| | FID score ↓ | PSNR ↑ | SSIM ↑ | LPIPS ↓ | MUSIQ score ↑ |
| GD Dhariwal & Nichol (2021) | 140.3 | 17.5 | 0.52 | 0.32 | 55.26 |
| GD + KC loss(Ours) | **125.5** | **18.7** | **0.56** | **0.27** | **57.33** |
| LD. Karras et al. (2022) | 103.2 | 18.7 | 0.59 | 0.25 | 58.62 |
| LD + KC loss(Ours) | **80.1** | **19.5** | **0.67** | **0.20** | **60.31** |

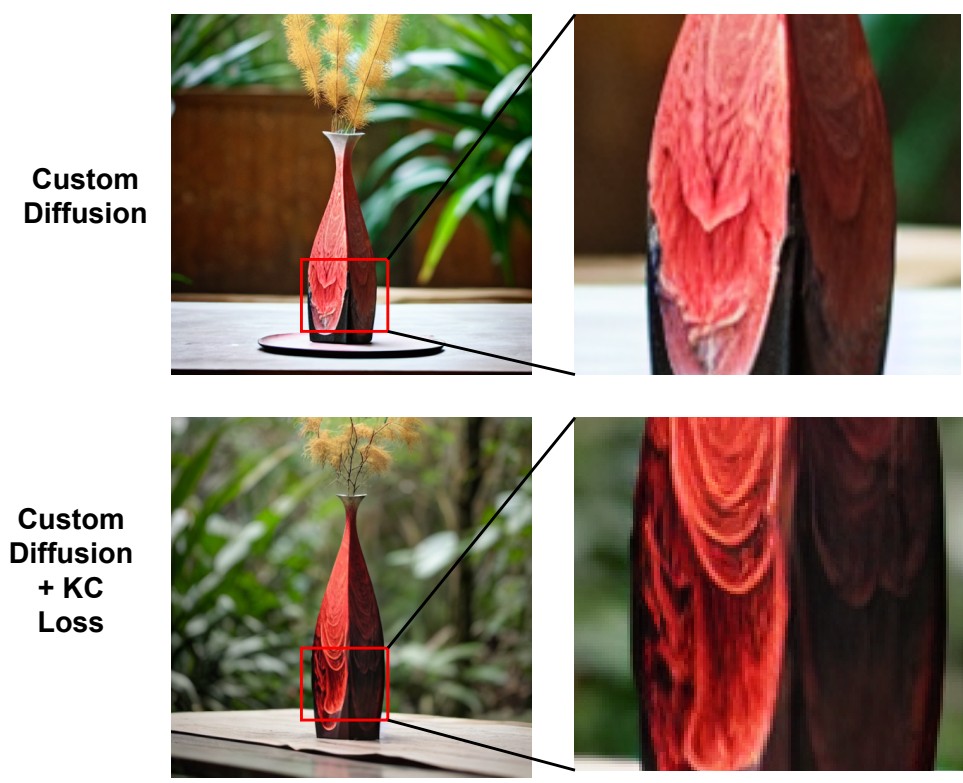

Figure 20: Qualitative comparison of with/without KC loss in Custom diffusion. The bottom image (with KC loss) shows better image quality in terms of color vividness and contrast (best viewed in color).

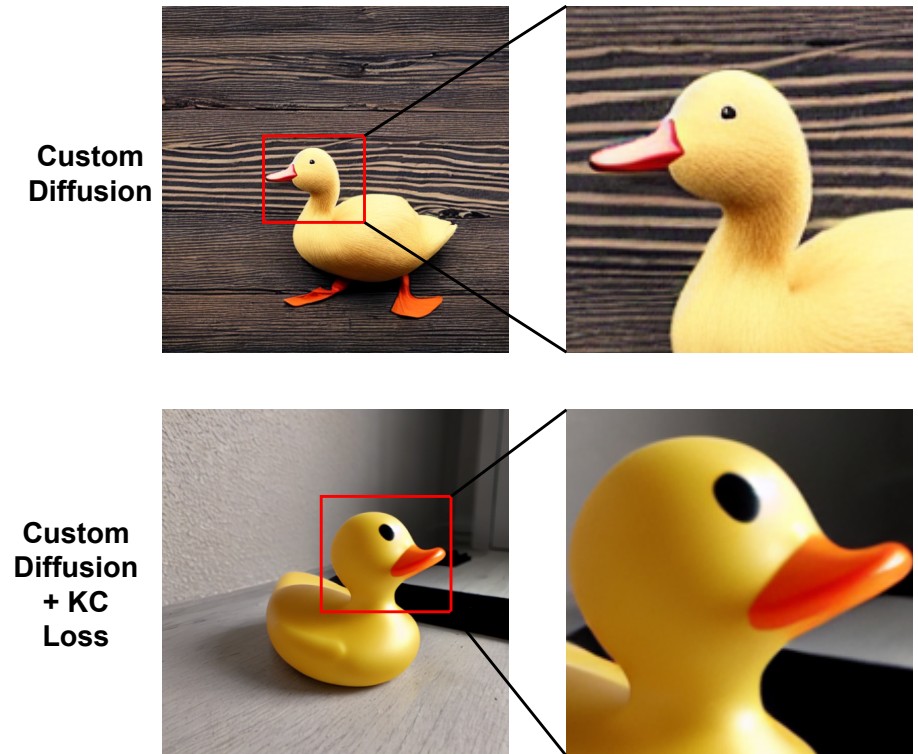

Figure 21: Qualitative comparison of with/without KC loss in Custom diffusion. The bottom image (with KC loss) shows better image quality in terms of detail and smoothness (best viewed in color).

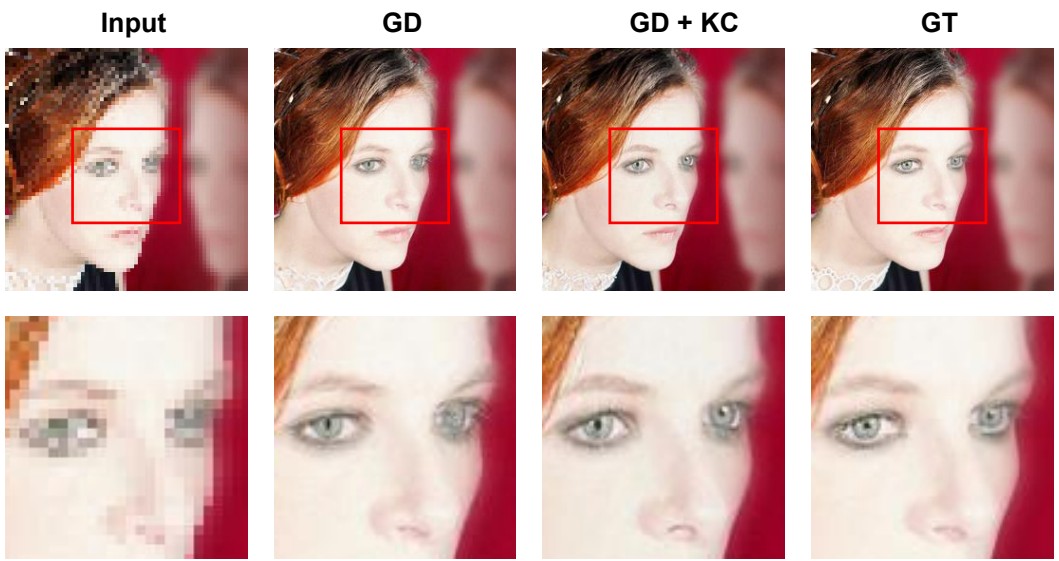

Figure 22: Qualitative comparison of with/without KC loss in guided diffusion (GD). The bottom image (with KC loss) has better eye and hair details (best viewed in color).

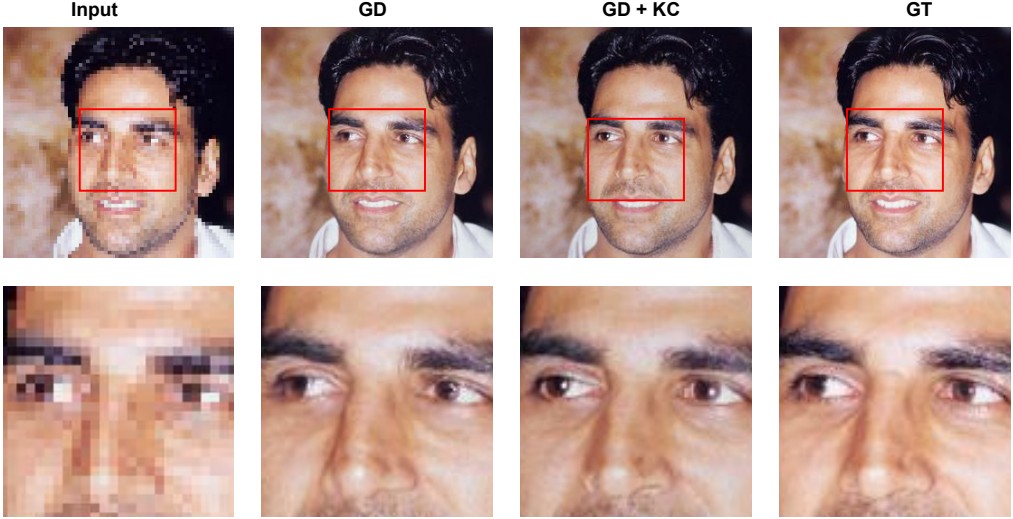

Figure 23: Qualitative comparison of with/without KC loss in guided diffusion (GD). The bottom image (with KC loss) has better eye details and skin smoothness (best viewed in color).

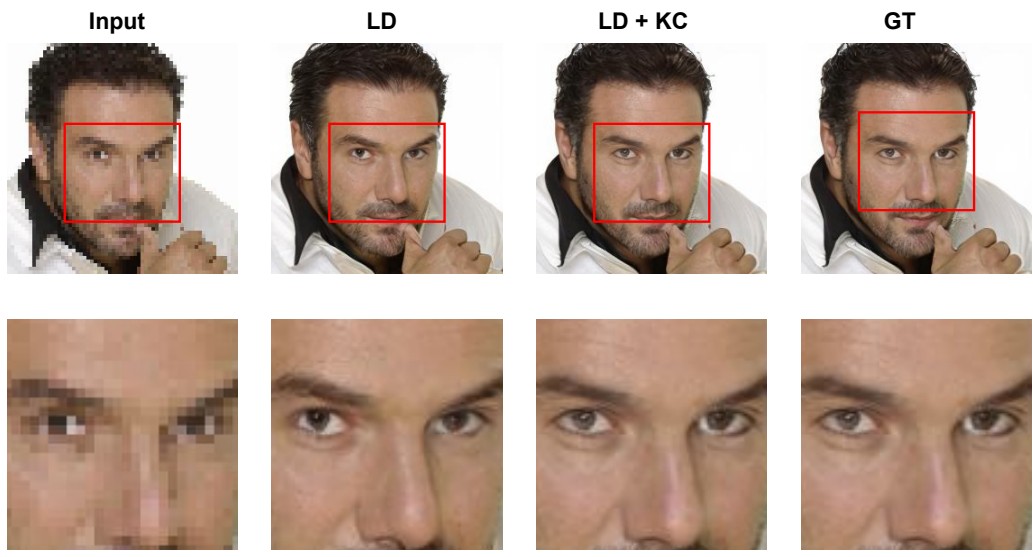

Figure 24: Qualitative comparison of with/without KC loss in Latent diffusion (LD). The bottom image (with KC loss) has higher similarity w.r.t the ground truth in terms of left eye and skin color (best viewed in color).

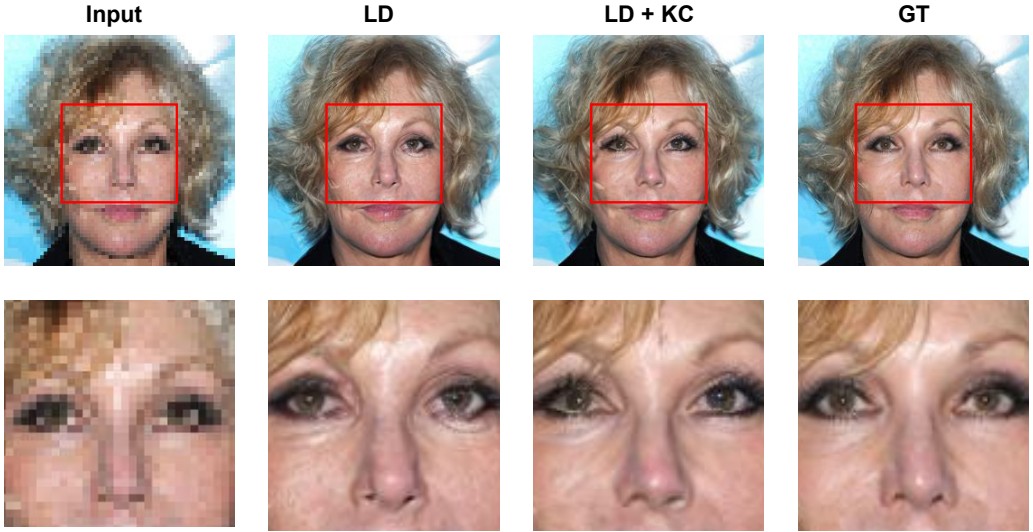

Figure 25: Qualitative comparison of with/without KC loss in Latent diffusion (LD). The bottom image (with KC loss) has better eye details and skin smoothness (best viewed in color).

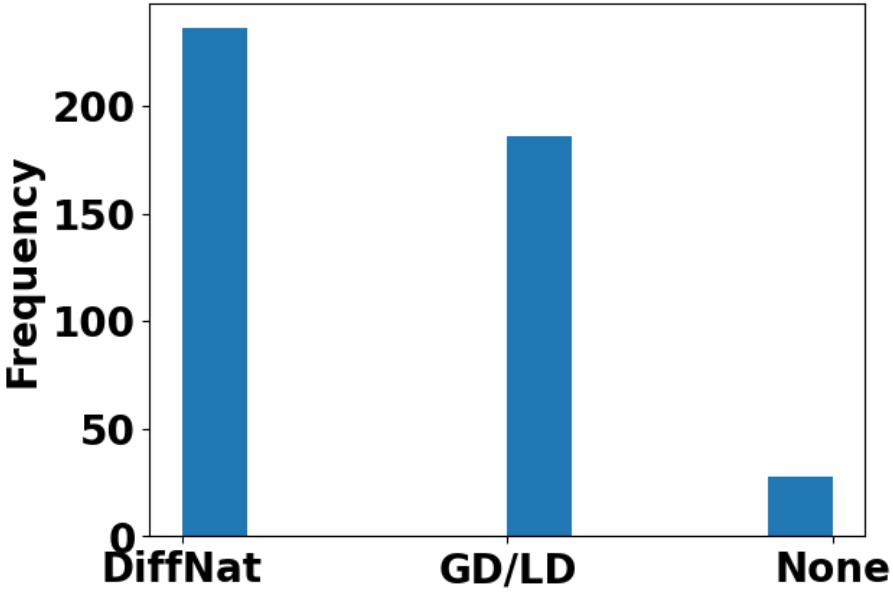

Figure 26: Human evaluation for image super-resolution task. DiffNat performs better than guided diffusion (GD), latent diffusion (LD) in user study as well.

