# OpenReview forum: "DIFFNAT: IMPROVING DIFFUSION IMAGE QUALITY USING NATURAL IMAGE STATISTICS"
_ICLR.cc/2024/Conference — Submitted to ICLR 2024_

### Official Review · Reviewer_NS8z · 2023-10-25

**Soundness:** 3 good
**Presentation:** 2 fair
**Contribution:** 3 good
**Rating:** 6
**Confidence:** 4

**Summary:**

This paper proposes a “naturalness” preserving loss function that can improve generative image quality on diverse tasks including image super-resolution,  unconditional image generation, and personalized few-shot fine-tuning. The proposed loss function is called the kurtosis concentration (KC) loss, which encourages the kurtosis values across different DWT (Discrete Wavelet Transform) of the images to be constant.

**Strengths:**

1. The proposed method for improving image quality is simple but effective and can be applied in a plug-and-play manner. The theory of KC loss has a solid mathematical base and is persuasive.

2. The quantitative experiments are sufficient to show that KC loss can effectively improve image generative quality.

**Weaknesses:**

1. The visual results in Fig 6 do not show evident superiority of using KC loss. It is recommended to provide visual results with their corresponding KC statistic maps.

2. The computational complexity of KC loss is not discussed. It is recommended to report the additional time consumption caused by KC loss.

**Questions:**

Please see the Weaknesses.

---

> ### Author Response · Authors · 2023-11-14
> **Response to Reviewer NS8z**
>
> **Q1. [The visual results in Fig 6 do not show evident superiority of using KC loss. It is recommended to provide more visual results]**
>
> Response: Thanks for the suggestion!  More visual results have been provided in Fig. 18, 19, 20, 21, 22, 23, 24, 25 (Appendix).
>
> **Q2. [The computational complexity of KC loss is not discussed. It is recommended to report the additional time consumption caused by KC loss.]**
>
> Response: The computation complexity of the proposed loss is presented in Sec. F (Computation complexity). The runtime analysis has been provided in Sec. G (Training time analysis) and Tab. 6 (Appendix).

---

> > ### Author Response · Authors · 2023-11-17
> > **Has our response addressed your concerns?**
> >
> > Dear Reviewer NS8z,
> >
> > We hope to hear your feedback after our responses. Please let us know if our replies have sufficiently addressed your concerns and whether you have further feeback for improvements. Thank you for your time and consideration.

---

> > > ### Comment · Reviewer_NS8z · 2023-11-21
> > >
> > > The author's rebuttal addresses most of my concerns, and I'll keep my score.

---

> > > > ### Author Response · Authors · 2023-11-21
> > > > **Thank you!**
> > > >
> > > > Dear Reviewer NS8z,
> > > >
> > > > Thank you for your feedback.
> > > >
> > > > Best,
> > > > Authors.

---

### Official Review · Reviewer_XKja · 2023-10-30

**Soundness:** 2 fair
**Presentation:** 3 good
**Contribution:** 2 fair
**Rating:** 6
**Confidence:** 3

**Summary:**

In this paper, the authors utilize a generic naturalness, i.e. the kurtosis concentration (KC) loss, to improve diffusion model. The authors validate the proposed KC loss on three diffusion-based tasks, the experimental results show that introducing appropriate prior of natural image could lead to better generation model.

**Strengths:**

1) The authors validated the effectiveness of KC loss and show that appropriate prior of natural image is beneficial for diffusion model training.
2) The proposed KC loss is easy to optimize.
3) The authors validated the proposed loss on three widely studied tasks, the superiority experimental results validated the effectiveness of the proposed method.

**Weaknesses:**

1) The authors argue that they provided theoretical insights into the advantage of KC loss, however, the theorems only analyzed the correctness of KC loss but did not analyze why the KC loss is beneficial for diffusion model training. I think the authors over claimed their contribution.
2) In the literature of natural image prior modeling, a large variaty of models have been suggested for modeling natural image prior. Besides KC loss, are there any other priors which is beneficial for training diffusion models.
3) The authors only analyzed the final generation quality with widely used image quality metrics, it will be the best if the authors could further present other properties of the proposed model, for example, can KC loss improve the convergence speed, will the additional KC loss leads to longer training time?

**Questions:**

Please refer to the weakness part.

---

> ### Author Response · Authors · 2023-11-14
> **Response to Reviewer XKja**
>
> **Q1. [the theorems only analyzed the correctness of KC loss but did not analyze why the KC loss is beneficial for diffusion model training]**
>
> Response: In the theoretical analysis, we motivate justifying how the KC loss improves denoising. Note that we are not proposing a new training methodology for diffusion models, rather the proposed loss is a plug-and-play module which can be integrated into any standard diffusion model pipeline. (as shown empirically in Sec. 4.1, 4.2, 4.3 of the main paper).
>
> **Q2. [ Besides KC loss, are there any other priors which are beneficial for training diffusion models?]**
>
> Response: The most well-known prior for natural images is scale invariance, and (Zhang & Lyu) show that kurtosis concentration property can be motivated by the well-known prior for natural images, characterized by scale invariance. There might be some other natural image priors, but we haven’t considered those in this paper (beyond the scope of this paper). Alternatively, we have also compared our method with LPIPS loss (i.e., a perceptual loss) , which we have added as a baseline as shown in Tab. 1, Tab. 2, Tab. 3 in the revised version.
>
> **Q3. [it will be the best if the authors could further present other properties of the proposed model, for example, can KC loss improve the convergence speed, will the additional KC loss leads to longer training time?]**
>
> Response: Thanks for the suggestion! We have provided the complexity analysis of the proposed KC loss in Sec. F (Computation complexity). Additional experiments for timing analysis have been provided in Table. 6 (training time analysis).
>
> The main idea of the diffusion model is to train a UNet, which learns to denoise from a random noise to a specific image distribution. More denoised steps typically ensure a better denoised version of the image, as seen in DDPM, LDM. In proposition 1, we show that minimizing projection kurtosis further denoise input signals. Therefore, KC loss helps in the denoising process and improves the convergence speed of the UNet. This has been discussed in the paragraph right after proposition 1 in the main paper. Fig. 3 shows using the same number of steps (400), a model trained with KC loss generates better quality images (measured in terms of SNR) than a standard guided diffusion model (both models were trained for the same number of epochs). This has been addressed in Sec. I (convergence analysis) in the Appendix of the updated version of the paper. The Loss convergence is shown in Fig. 14 in the Appendix of the updated version of the paper.

---

> > ### Author Response · Authors · 2023-11-17
> > **Has our response addressed your concerns?**
> >
> > Dear Reviewer XKja,
> >
> > We hope to hear your feedback after our responses. Please let us know if our replies have sufficiently addressed your concerns and whether you have further feeback for improvements. Thank you for your time and consideration.

---

> > > ### Author Response · Authors · 2023-11-21
> > > **Has our response addressed your concerns?**
> > >
> > > Dear Reviewer XKja,
> > >
> > > Thank you for your time and effort. **As you know the author-reviewer discussion period ends tomorrow.** Please let us know if our rebuttal has sufficiently addressed your concerns and whether you have further questions for improvements.
> > >
> > > Best,
> > > authors.

---

### Official Review · Reviewer_4XiE · 2023-10-30

**Soundness:** 3 good
**Presentation:** 3 good
**Contribution:** 3 good
**Rating:** 5
**Confidence:** 2

**Summary:**

This paper presents a novel kurtosis concentration loss to preserve image "naturalness" to enhance image quality within standard diffusion model pipelines by reducing the kurtosis gap between band-pass image versions. The evaluation of the KC loss across diverse tasks show consistent improvements in perceptual quality via metrics like FID, MUSIQ score, and user evaluations.

**Strengths:**

1. The idea is interesting and novel. The idea of designing a quality measure to align the distribution of natural images and generated images from the diffusion models is novel.
2. The paper is clearly written. The reviewers can easily understand what the authors hope to convey.

**Weaknesses:**

1. The experiments are not extensive. More comparisons on the conditional image-to-image translation should be provided.
2. Despite improvement in quantitative results, the visual differences in using the loss or not are minor.
3. This technical route, i.e. using IQA for improving generation quality, seems to be less effective than other alternative routes, e.g. [1].
[1] Susung Hong, Gyuseong Lee, Wooseok Jang, Seungryong Kim, "Improving Sample Quality of Diffusion Models Using Self-Attention Guidance," ICCV, 2023.
4. The presented visual results are in small resolutions. Higher resolution results are suggested for putting into the paper.

**Questions:**

1. Why are the FID values in the paper so high compared to those in other papers?
2. The PSNR results shown in Table 3 are also very low.

---

> ### Author Response · Authors · 2023-11-14
> **Response to Reviewer 4XiE**
>
> **Q1. [The experiments are not extensive. More comparisons on the conditional image-to-image translation should be provided.]**
>
> Response: We have updated the main paper with additional experiments using more baselines in Tab. 1, Tab. 2, Tab. 3. We also include additional qualitative results in Fig. 18, 19, 20, 21, 22, 23, 24, 25 in the Appendix.
>
>
> **Q2. [Despite improvement in quantitative results, the visual differences in using the loss or not are minor.]**
>
> Response: We have provided more qualitative results (zoomed version) in Fig. 18, 19, 20, 21, 22, 23, 24, 25 (Appendix).
>
>
> **Q3. [less effective than other alternative routes, e.g. [1]. [1] Susung Hong, Gyuseong Lee, Wooseok Jang, Seungryong Kim, "Improving Sample Quality of Diffusion Models Using Self-Attention Guidance," ICCV, 2023.]**
>
> Response: Our proposed loss function is used during training diffusion models. On the other hand, [1] is a training free approach and they propose to use self-attention guidance during inference. Therefore, these approaches are not comparable, but rather complementary to each other. Moreover, [1] uses an additional forward pass to generate the attention map, which increases the computation complexity 2x compared to guidance free method, whereas our method introduces very less overhead during training (See Table. 6, Appendix).
>
> **Q4. [The presented visual results are in small resolutions. Higher resolution results are suggested for putting into the paper.]**
>
> Response: We have presented the results in Fig. 11, 12, 18, 19, 20, 21, 22, 23, 24, 25 (Appendix) in high resolution.
>
> **Q5. [Why are the FID values in the paper so high compared to those in other papers? The PSNR results shown in Table 3 are also very low.]**
>
> Response: For a fair comparison, we have reproduced the baselines with the same training setting and images, and the results are reported. In a similar setting, using KC loss produces images with lesser FID and higher MUSIQ scores across tasks and datasets, which validate the efficacy of the proposed KC loss.

---

> > ### Author Response · Authors · 2023-11-17
> > **Has our response addressed your concerns?**
> >
> > Dear Reviewer 4XiE,
> >
> > We hope to hear your feedback after our responses.
> > We have incorporated additional experiments for the **image super-resolution task** and updated the paper. Quantitative results on image super-resolution for **x2** and **x8** tasks have been added in **Sec. K** (Experiments on image super-resolution, Tab. 7 and Tab. 8 in Appendix). We have also performed human evaluation for this task in **Sec K.2** (human evaluation, Fig. 26 in Appendix). **All changes are highlighed in blue in the updated version of the paper.**
> > Please let us know if our replies have sufficiently addressed your concerns and whether you have further feeback for improvements. Thank you for your time and consideration.

---

> > > ### Author Response · Authors · 2023-11-21
> > > **Has our response addressed your concerns?**
> > >
> > > Dear Reviewer 4XiE,
> > >
> > > Thank you for your time and effort. **As you know the author-reviewer discussion period ends tomorrow.** Please let us know if our rebuttal has sufficiently addressed your concerns and whether you have further questions for improvements.
> > >
> > > Best,
> > > authors.

---

### Official Review · Reviewer_en6K · 2023-10-31

**Soundness:** 2 fair
**Presentation:** 3 good
**Contribution:** 2 fair
**Rating:** 5
**Confidence:** 4

**Summary:**

This paper proposes to improve quality and reduce artifacts in diffusion-based generative models. More specifically, they propose a kurtosis concentration loss to increase SNR and therefore improve results' quality. Their method can be applied a wide range of generative pipelines.

**Strengths:**

1. I think the mentioned problem of naturalness in generative process is important. And the idea to use natural image priors to improve quality is reasonable.

2. The paper writing is clear to follow.

**Weaknesses:**

1. My main concern is about effectiveness of this KC loss. In the experiment part, the authors gives FID and MUSIQ in Table 1, 2, which I think shows little improvements. And we all know that these metrics can not accurately measure image quality.
Moreover, in Fig 6, Fig 8, Fig 9 and Fig 10, it is not obvious whether KC loss yields better results. I think the authors need to at lease hightlight the region or illustrate using zoom-out regions to show the difference.

2. Although I also appreciate very much those early works about natural image priors, I think most of them are based on simplistic assumptions which may not be needed in today's large-data-driven methods.
More specically, I doubt that why kurtosis-based losses suit for the mentioned artifacts problem? Actually, losses need to be designed based on properties that you want to distinguish. I think at least statistics of a large amount of artifact images are needed. to prove kurtosis is a proper metric.

3. In Fig 1, the authors show severe unnatural artifacts. However, I wonder is that really a common phenomenon? It is more like a bug in training process, and similar artifacts are not present in the other illustrations in the paper. Then what exactly are the types of artifacts that the authors want to solve?

4. In Equ 5, the authors claim that diffusion models are typically trained using reconstruction loss in image domain. However, as far as I known, many diffusion models do not use such reconstruction loss, for example DDPM, DDIM.
Moreover, the loss and optimization of diffusion model typically follow mathematical derivations. I wonder if we incorporate KC loss, do the training process still theoretically sound?

**Questions:**

Please address the problems in weaknessed part.

---

> ### Author Response · Authors · 2023-11-14
> **Response to reviewer en6K**
>
> **Q1. [My main concern is about the effectiveness of this KC loss ..authors need to at least highlight the region or illustrate using zoom-out regions to show the difference]**
>
> Response: The effectiveness of KC loss has been shown quantitatively in Tables 1, 2, 3 (main paper). We have shown some qualitative examples in the paper, and based on the suggestion we have added more results highlighting the improved image quality. We have also verified the quality improvement using user study (Sec. human evaluation, page 7 main paper) by Amazon Mechanical Turks, and show that the generated images are indeed high quality. The comprehensive analysis is added in Sec J (Qualitative analysis, Appendix) ( Fig. 18, 19, 20, 21, 22, 23, 24, 25 in Appendix).
>
> **Q2. [Although I also appreciate very much those early works about natural image priors, I think most of them are based on simplistic assumptions which may not be needed in today's large-data-driven methods]**
>
> Response: We use the naturalness prior to design a plug-and-play loss function to improve the image quality of the generated images. The loss can be added to any diffusion pipeline and further improve the image quality which is experimentally validated in the paper (Tab. 1, Tab. 2, Tab. 3 in the main paper). Hence, we contend that this loss does not serve as a substitute for large-scale data-driven methods; instead, it complements and enhances existing pipelines.
>
> **Q3. [I doubt that why kurtosis-based losses suit for the mentioned artifacts problem?.. at least statistics of a large amount of artifact images are needed. to prove kurtosis is a proper metric]**
>
> Response: We would also like to clarify that the artifact issue is limited to some particular approaches (personalized few-shot finetuning, e.g., dreambooth, custom diffusion). Overall, our goal is to improve the image quality. More experimental results are added to visually show this in  Fig. 18, 19, 20, 21, 22, 23, 24, 25 (Appendix).
>
> We also provide the average kurtosis analysis (Sec. H ) in the Appendix of the updated paper. It is observed that integrating KC loss reduces the kurtosis deviation, which resembles average natural image kurtosis statistics (Fig. 15, Fig. 16, Fig. 17 in the Appendix).
>
> **Q4. [In Fig 1, the authors show severe unnatural artifacts. However, I wonder is that really a common phenomenon?.. Then what exactly are the types of artifacts that the authors want to solve?]**
>
> Response: Our primary goal is to improve the image quality of generated images produced by diffusion models for different tasks. We are not specifically targeting any unnatural artifact reduction during the generation process. One of the tasks is personalized few-shot finetuning (e.g., Dreambooth), featured in the teaser figure (Fig. 1). In Dreambooth, having unnatural artifacts is a known phenomenon (Ruiz et al.) since overfitting/underfitting might happen due to training a large model from a few examples, which might lead to poor-quality images, having higher FID scores. Therefore, the unnatural artifact issue is just an anecdotal instance and is not the primary motive of the paper. For example, this is not an issue in unconditional image generation, image super-resolution tasks. We primarily focus on improving the overall image quality in this paper.

---

> > ### Author Response · Authors · 2023-11-14
> > **More comments**
> >
> > **Q5. [In Equ 5, the authors claim that diffusion models are typically trained using reconstruction loss in the image domain. However, as far as I know, many diffusion models do not use such reconstruction loss, for example DDPM, DDIM. ]**
> >
> > Response: We apologize for the confusion. We are proposing a generic loss function inspired by the kurtosis concentration property, which can be applied to several diffusion-based tasks. By reconstruction loss, we mean the widely used simple diffusion loss ($L_{simple}$) as mentioned in (Rombach et al.), which is the L2 difference between the ground truth image and output of the diffusion UNet, which takes either the noisy image or its latent counterpart as the input.  We have clarified this in the updated version of the paper (marked as blue in page 5, main paper).  To be more generic, we have also mentioned task-specific loss ($L_{task}$), which can vary for specific tasks, e.g., Dreambooth has (1) reconstruction loss and (2) prior preserving loss.
> >
> > **Q6. [Moreover, the loss and optimization of diffusion models typically follow mathematical derivations. I wonder if we incorporate KC loss, does the training process still theoretically sound?]**
> >
> > Response: The main idea of the diffusion model is to train a UNet, which learns to denoise from a random noise to a specific image distribution. More denoised steps typically ensure a better denoised version of the image, as seen in DDPM, LDM. In proposition 1, we show that minimizing projection kurtosis further denoise input signals. Therefore, KC loss helps in the denoising process and improves the convergence speed of the UNet. This has been discussed in the paragraph right after proposition 1 in the main paper. Fig. 3 shows using the same number of steps (400), a model trained with KC loss generates better quality images (measured in terms of SNR) than a standard guided diffusion model (both models were trained for the same number of epochs). This has been addressed in Sec. I (convergence analysis) in the Appendix of the updated version of the paper. The loss convergence is shown in Fig. 14 in the Appendix of the updated version of the paper.
> >
> > Note that we are not proposing a new training methodology for diffusion models, rather the proposed loss is a plug-and-play module which can be integrated into any standard diffusion model pipeline.

---

> > > ### Author Response · Authors · 2023-11-17
> > > **Has our response addressed your concerns?**
> > >
> > > Dear Reviewer en6K,
> > >
> > > We hope to hear your feedback after our responses. Please let us know if our replies have sufficiently addressed your concerns and whether you have further feeback for improvements. Thank you for your time and consideration.

---

> > > > ### Comment · Reviewer_en6K · 2023-11-19
> > > > **Thanks for the response**
> > > >
> > > > Thanks for the response for previous comments. I read the response and the revised paper.
> > > > The response partly addressed my questions. But I think several questions are still not satisfying.
> > > >
> > > > 1. `Artifact` / 'unnatural' is too general. Previously I want to know what exacly the kind of artifacts that this paper want to address. Color saturation, noise, blocking or others? And it would be better to explain a little more why KC loss can solve these.
> > > >
> > > > 2. In Q3, I mean do those artifacts images really have larger KC values on statistics over a large dataset?
> > > > In Fig 17, the model with KC loss minimization will definitely has lower Kurtosis. But why `natural images have more concentrated kurtosis values'? I think the authors explain the effectiveness of the proposed method based on others' assumption, neglecting on what situation, for what kind of artifacts, on what kind of data (face or senary?) where the assumption works.
> > > >
> > > > 3. In Q6, I mean the orignal loss and model of diffusion follows strict mathematical derivations. I just wonder if use KC loss, can we still find a suitable and strict derivations for that? However, I think the authors misunderstand and have not think in this direction. Indeed, neural networks can add loss at will and does not necessarily need to be strictly derived.
> > > >
> > > > I like the idea to use priors for networks. However, I think we need to know very clearly what exact problem we want to solve and why these priors are suitable and theorectically correct, rather than just borrow and use like a black box. Therefore, I think I still lean to reject.

---

> > > > > ### Author Response · Authors · 2023-11-20
> > > > > **Response to Reviewer en6K (Part 1)**
> > > > >
> > > > > We thank the reviewer for their insightful comments. The response is provided below.
> > > > >
> > > > > **Q1. ['unnatural' is too general]**
> > > > >
> > > > > Ans: We are not addressing any specific type of artifacts, rather, we aim to improve the overall quality of the generated images through the lens of kurtosis-based statistical properties. Similarly, for validation, we resort to standard metrics like the FID, MUSIQ score, which measures the similarity between the overall distribution of the generated images and the real ones, rather than relying on selectively chosen properties.
> > > > >
> > > > > We admit ''naturalness'' is a generic term for image quality, and therefore, image quality metrics e.g., SSIM [7], BRISQUE [8] etc. are computed considering luminance, chrominance, contrast, structure, color in the image all together. We are not specifically addressing any particular kind of artifacts and not claiming that those artifacts could be mitigated through KC loss. That would have been more particular and infeasible, given the ''naturalness'' or image quality evaluation itself is a complex phenomenon. Instead, we are exploiting the KC property to propose KC loss which can be added to any standard diffusion pipeline as a plug-and-play component in order to preserve the ''naturalness'' and ameliorate image quality of the generated images. Therefore, KC loss serves as cause and the effect would be generating better quality images.
> > > > > We experimentally validate this through quantitative (FID, MUSIQ score comparison in Tab. 1, 2, 3 main paper), qualitative analysis (Fig. 6, Fig. 8, Fig. 9, Fig. 10 main paper) as well as user studies (Sec. human evaluation) for three diverse tasks with multiple appraches. We empirically observe better image quality in terms of absense artifacts in the edges (Fig. 18, Appendix), improved contrast and reflections (Fig. 19, Appendix), better color consistency (Fig. 20, Appendix), smoother texture (in Fig. 21, Appendix), better eye structure preservation (Fig. 22, Appendix), better eye details and skin smoothness (Fig. 23, 24, 25 Appendix) while adding KC loss. We will clarify this in the updated version of the paper.

---

> > > > > > ### Author Response · Authors · 2023-11-20
> > > > > > **Response to Reviewer en6K (Part 2)**
> > > > > >
> > > > > > **Q2. [But why 'natural images have more concentrated kurtosis values'?]**
> > > > > >
> > > > > > Ans: The primary motivation of the paper stems from the well-known kurtosis concentration property of natural images, which is well-established both theoretically and experimentally [1, 2, 3].
> > > > > >
> > > > > > It is well-known that natural images can be modeled using zero-mean GSM vector [2, 4, 7, 10, 11, 12], and based on this assumption theoretically it is proved that projection kurtosis of bandpass filtered version of the images are constant (Lemma 1, main paper). We experimentally verify this property for natural images for large datasets, e.g., Dreambooth dataset (Fig. 15, Appendix) , Oxford-flowers dataset (Fig. 16, Appendix), FFHQ dataset (Fig. 17, Appendix). Therefore, this property holds for both object datasets (Dreambooth dataset, Oxford flowers), face dataset (FFHQ). This is reasonable since the necessary conditions for this assumption to hold is the input to be an zero-mean Gaussian Scale Mixture model, which natural images generally holds.
> > > > > >
> > > > > > Then, we use this property as a prior to the standard diffusion model pipeline to apply this constrain in order to generate images of better quality. We have observed that KC loss improves image quality quantitatively, qualitatively and using user study for three tasks with multiple approaches. This validates KC loss to be an useful prior in multiple settings across different datasets.
> > > > > >
> > > > > >
> > > > > > One intuitive reasoning of the projected kurtosis concentration property is given as follows - different bandpass (DWT) filtered versions of natural images generally have a generalized Gaussian density of the following form [2, 5].
> > > > > > \begin{equation}
> > > > > > p(x, \alpha, \beta) = \frac{\beta}{2\alpha \Gamma(1/\beta)} exp(-\frac{|x|}{\alpha})^{\beta}
> > > > > > \end{equation}
> > > > > >
> > > > > > The kurtosis of this function is given by [5],
> > > > > > \begin{equation}
> > > > > > \kappa = \frac{\Gamma(1/\beta) \Gamma(5/\beta)}{\Gamma(3/\beta)^2}
> > > > > > \end{equation}
> > > > > >
> > > > > > Empirically, it has been shown that for natual images, $\beta$ is relatively small values ranges from 0.5 to 1 [2, 6], and this kurtosis value tend to be constant [1, 2, 3, 6], independent of $\alpha$ or $x$.
> > > > > >
> > > > > > Ref:
> > > > > >
> > > > > > [1] Xing Zhang and Siwei Lyu. Using projection kurtosis concentration of natural images for blind noise covariance matrix estimation. In Proceedings of the IEEE Conference on Computer Vision and Pattern Recognition, pp. 2870–2876, 2014.
> > > > > >
> > > > > > [2] Daniel Zoran and Yair Weiss. Scale invariance and noise in natural images. In 2009 IEEE 12th International Conference on Computer Vision, pp. 2209–2216. IEEE, 2009.
> > > > > >
> > > > > > [3] Lyu S, Pan X, Zhang X. Exposing region splicing forgeries with blind local noise estimation. International journal of computer vision. 2014 Nov;110:202-21.
> > > > > >
> > > > > > [4] Wainwright MJ, Simoncelli E. Scale mixtures of Gaussians and the statistics of natural images. Advances in neural information processing systems. 1999;12.
> > > > > >
> > > > > > [5] Buccigrossi, Robert W., and Eero P. Simoncelli. "Image compression via joint statistical characterization in the wavelet domain." IEEE transactions on Image processing 8.12 (1999): 1688-1701.
> > > > > >
> > > > > > [6] A. Srivastava, A. B. Lee, E. P. Simoncelli, and S.-C. Zhu. On advances in statistical modeling of natural images. J. Math. Imaging Vis., 18(1):17–33, 2003.
> > > > > >
> > > > > > [7] Ruderman, Daniel L., and William Bialek. "Statistics of natural images: Scaling in the woods." Physical review letters 73.6 (1994): 814.
> > > > > >
> > > > > > [8] Wang, Zhou; Bovik, A.C.; Sheikh, H.R.; Simoncelli, E.P. (2004-04-01). "Image quality assessment: from error visibility to structural similarity". IEEE Transactions on Image Processing. 13 (4): 600–612.
> > > > > >
> > > > > > [9] Mittal, Anish, Anush Krishna Moorthy, and Alan Conrad Bovik. "No-reference image quality assessment in the spatial domain." IEEE Transactions on image processing 21.12 (2012): 4695-4708.
> > > > > >
> > > > > > [10] Mumford, David Bryant. "Empirical statistics and stochastic models for visual signals." (2006).
> > > > > >
> > > > > > [11] D. Mumford and B. Gidas, “Stochastic Models for Generic Images,” Quarterly J. Applied Math., vol. LIX, no. 1, pp. 85-111, 2001.
> > > > > >
> > > > > > [12] D. Field, “Relations between the Statistics of Natural Images and the Response Properties of Cortial Cells,” J. Optical Soc. Am. A, vol. 4, no. 12, pp. 2379-2394, Dec. 1987

---

> > > > > > > ### Author Response · Authors · 2023-11-20
> > > > > > > **Response to Reviewer en6K (Part 3)**
> > > > > > >
> > > > > > > **Q3. [I just wonder if use KC loss, can we still find a suitable and strict derivations for that?]**
> > > > > > >
> > > > > > > Ans: We would like to highlight that in our work, the underlying theoretical framework behind the forward and reverse diffusion processes remains unchanged; rather, we focus on improving the performance of the denoising neural network used to approximate the reverse diffusion trajectory.
> > > > > > >
> > > > > > > Suppose, we have the input training images ($\{x\}$) and conditioning vector $c$. The conditioning vector could be text (text-to-image model), image (image-to-image model), or none (in case of the unconditional diffusion model). In the forward process [13, 14], the noisy versions of image $x$ at timestep $t$ is generated as $x_t = \alpha_{t} x +\sigma_{t} \epsilon$, where $\epsilon \sim N(0,I)$.
> > > > > > >
> > > > > > > In the reverse process [13, 14], a denoised autoencoder (${f_{\theta}}$) is trained to predict the denoised version of the image ($x_{t, gen}$) at each timestep $t$ from the noisy images $x_t$, i.e., $x_{t, gen} = f_{\theta} (x_t, c, t)$.
> > > > > > > Typically, the denoised autoencoder (${f_{\theta}}$) is trained by minimizing the Mean Squared Error between the real image ($x$) and the generated denoised version of the image at time step $t$ ($x_{t, gen}$) averaged over timesteps and noise variances as denoted by,
> > > > > > >
> > > > > > > \begin{equation}
> > > > > > > L_{recon} = E_{x,c, \epsilon, t} [ \ || x_{t, gen} - x ||_{2}^{2}]
> > > > > > > \end{equation}
> > > > > > >
> > > > > > >
> > > > > > > The kurtosis concentration loss is applied on the generated images ($x_{t, gen}$), and therefore can be considered as a function ($f'$) of $x_{gen}$ as follows:
> > > > > > >
> > > > > > > \begin{equation}
> > > > > > > L_{KC} = E_{x, c, \epsilon, t} [ f'( x_{t, gen}) ]
> > > > > > > \end{equation}
> > > > > > >
> > > > > > > Note the function $f'$ is difference between the maximum and minimum values of the DWT filtered version of input $x_{t, gen}$.
> > > > > > >
> > > > > > > Therefore, the total loss function can be written as,
> > > > > > > \begin{equation}
> > > > > > > L_{total} = E_{x,c, \epsilon, t} [\||x_{t, gen} - x\||^2_2 +  f'( x_{t, gen})]
> > > > > > > \end{equation}
> > > > > > >
> > > > > > > In our work, the above-mentioned framework remains the same. Instead, the proposed KC loss acts as an additional regularizer to the training of the denoising neural network, which helps it to denoise $x_t$ better (Lemma 2, main paper), ultimately improving the approximation of $x$, i.e., $x_{t, gen}$ at each time step t.
> > > > > > >
> > > > > > > Ref:
> > > > > > >
> > > > > > > [13] Ruiz, Nataniel, et al. "Dreambooth: Fine tuning text-to-image diffusion models for subject-driven generation." Proceedings of the IEEE/CVF Conference on Computer Vision and Pattern Recognition. 2023.
> > > > > > >
> > > > > > > [14] Ho, Jonathan, Ajay Jain, and Pieter Abbeel. "Denoising diffusion probabilistic models." Advances in neural information processing systems 33 (2020): 6840-6851.
> > > > > > >
> > > > > > > **Q4. [I like the idea to use priors for networks.]**
> > > > > > >
> > > > > > > Ans: We thank the reviewer for appreciating our idea.
> > > > > > >
> > > > > > > **Q5. [However, I think we need to know very clearly what exact problem we want to solve and why these priors are suitable and theoretically correct, rather than just borrow and use like a black box.]**
> > > > > > >
> > > > > > > Ans: We respectfully beg to differ with the reviewer on this point. Our exact problem is very clear, i.e., improving image quality of diffusion generated images, which is of paramount importance for diffusion tasks. We are not particularly addressing any specific artifacts, rather we aim to improve the overall image quality.
> > > > > > > Before proposing the loss function, we verify that projected kurtosis concentration property actually holds for natural images in large datasets, e.g., Dreambooth dataset (Fig. 15, Appendix), Oxford flower dataset (Fig. 16, Appendix), FFHQ dataset (Fig. 17, Appendix), encompassing diverse range of object, face datasets.
> > > > > > > Next, using KC prior we propose the KC loss, which has been motivated theoretically, and validated experimentally for diverse diffusion tasks across different datasets, not just a black box.

---

> > > > > > > > ### Author Response · Authors · 2023-11-21
> > > > > > > > **Response to Reviewer en6K**
> > > > > > > >
> > > > > > > > Dear Reviewer en6K,
> > > > > > > >
> > > > > > > > Thank you for your time and effort. **As you know the author-reviewer discussion period ends tomorrow.** Please let us know if our rebuttal has sufficiently addressed your concerns and whether you have further questions for improvements.
> > > > > > > >
> > > > > > > > Best,
> > > > > > > > authors.

---

### Author Response · Authors · 2023-11-14
**Response to reviewers**

**General comments**

We thank the reviewers for their useful comments. All the reviewers have appreciated the novelty of the approach and experimental validation. Their concerns are addressed below. We have updated the paper and changes are highlighted in blue. In the updated version of the paper, we have incorporated the following changes:

**1.** Added experiments and analysis for computation complexity

**2.** Training time analysis

**3.** Kurtosis analysis

**4.** Convergence analysis

**5.** Qualitative analysis

**6.** Baseline experiments

**7.** Clarify some details in the main paper.

The positive points identified by the reviewers are highlighted below.

**Positive points:**

**1. Novelty: Interesting and novel idea** [en6K, 4XiE, XKja, NS8z]

**2. Well-written:  easy to follow** [en6K, 4XiE]

**3. Extensive experiments: Strong results with multiple tasks** [XKja, NS8z]

---

> ### Author Response · Authors · 2023-11-17
> **General response to all reviewers and AC**
>
> We thank the AC for initiating the discussion.
>
> We have incorporated additional experiments for the **image super-resolution task** and updated the paper. Quantitative results on image super-resolution for **x2** and **x8** tasks have been added in **Sec. K** (Experiments on image super-resolution, Tab. 7 and Tab. 8 in Appendix). We have also performed human evaluation for this task in **Sec K.2** (human evaluation, Fig. 26 in Appendix). **All changes are highlighed in blue in the updated version of the paper.** We hope to hear from the reviewers.

---

### Comment · Area_Chair_3RN1 · 2023-11-15
**Please engage in reviewer-author discussion**

Dear reviewers,

The paper got diverging scores. The authors have provided their response to the comments. Could you look through the other reviews and engage into the discussion with authors? See if their response changes your assessment of the submission?

Thanks!
AC

---

### Author Response · Authors · 2023-11-23
**Final general response to reviewers and AC (Part 1)**

Dear AC and all the reviewers,

We appreciate your time and effort in reviewing our paper. We summarize the positive points identified by all the reviewers. **We addressed the concerns of the reviewers in detail and hope to get positive feeback from them (4XiE, XKja) soon since the rebuttal period is coming to an end.** For their convenience, we also add the summary of the experiments added in the updated verison of the paper. **These changes has been highlighed in blue in the final updated verison.**

**Positive points:**

**1. Novelty: Interesting and novel idea [en6K, 4XiE, XKja, NS8z]**
**2. Well-written: easy to follow [en6K, 4XiE]**
**3. Extensive experiments: Strong results with multiple tasks [XKja, NS8z]**


**Added experiments and analysis:**

**1. Computation complexity**
We analyze the computational complexity of the proposed KC loss. Suppose, given a batch of N images. We need to perform DWT of each images using k different filters. Since, DWT for 'haar' wavelet can be done in linear time, the complexity of performing DWT with k filters can be done in $\mathcal{O}(Nk)$ time. Now, calculating the difference between maximum and minimum kurtosis can be done in linear time, therefore, the computational complexity of calculating KC loss is $\mathcal{O}(Nk)$. This minimal overhead of computing KC loss can be observed in the training time analysis. This is included in **Sec. F (Appendix).**

**2. Training time analysis**
The run time analysis has been provided in Table. 1 . Note that the experiments for Dreambooth, Custom diffusion, DDPM have been performed on a single A5000 machine with 24GB GPU. We have performed guided diffusion (GD) and latent diffusion (LD) experiments on a server of 8 24GB A5000 GPUs. The experimental results show that incorporating KC loss induces minimum training overhead. This is included in **Sec. G (Appendix).**

**Table 1: Training time analysis**

|  Method      | dataset  | Training time |
|--------------|-------------|------------|
| DreamBooth [1] | 5-shot finetune DreamBooth dataset | 10m 21s |
| DreamBooth [1] + KC loss | 5-shot finetune DreamBooth dataset | 11m 30s |
| Custom diff [2] | 5-shot finetune DreamBooth dataset | 6m 43s|
| Custom diff [2] + KC loss | 5-shot finetune DreamBooth dataset | 7m 11s|
| DDPM [4] | CelebAfaces | 2d 8h 21 m |
| DDPM [4] + KC loss | CelebAfaces | 2d 9h 19 m |
| DDPM [4] | CelebAHQ | 21h 48m |
| DDPM [4] + KC loss | CelebAHQ | 22h 40m |
| DDPM [4] | Oxford flowers | 6h 17m |
| DDPM [4] + KC loss | Oxford flowers | 6h 39m |
| GD [5] | FFHQ | 23h 10m |
| GD [5] + KC loss | FFHQ | 1d 1h 29m |
| LD [6] | FFHQ | 20h 15m |
| LD [6] + KC loss | FFHQ | 22h 40m |

**3. Kurtosis analysis**

To verify the efficacy of the proposed KC loss, we perform average kurtosis analysis in **Sec. H (Appendix)**. We compute the average kurtosis deviation of DWT filtered version of images from the dataset and plot them in Fig. 15, 16, 17 (Appendix) in the updated version of the paper for Dreambooth dataset, Oxford flowers and FFHQ dataset.
All the plots show that adding KC loss minimizes the deviation of kurtosis values in the generated images, and natural images have least kurtosis deviation as conjectured by kurtosis concentration property.
This analysis verifies minimizing kurtosis loss improves diffusion image quality.


**4. Convergence analysis**

The main idea of the diffusion model is to train a UNet, which learns to denoise from a random noise to a specific image distribution. More denoising steps ensure a better denoised version of the image, e.g., DDPM [4], LDM [6]. In proposition 1 (main paper), we show that minimizing projection kurtosis further denoise input signals. Therefore, KC loss helps in the denoising process and improves the convergence speed. We have shown that adding KC loss improves the loss to converge faster for Dreambooth task in **Fig. 14 (Appendix)**. This is discussed in **Sec. I (Appendix)** of the updated verison of paper.

**5. Qualitative analysis**

In this section, we provide more qualitative analysis to show that adding KC loss improves image quality. Zoomed view of the generated images are shown to compare w.r.t the baselines in **Fig. 18, 19, 20, 21, 22, 23, 24, 25 (Appendix)**.

---

> ### Author Response · Authors · 2023-11-23
> **Final general response to reviewers and AC (Part 2)**
>
> **6. Quantitative analysis**
>
> More expermients with new baslines and additional settings are provided here. Experiments of personalized few-shot finetuning task have been provided in Table 2 (Tab 1 in main paper). Unconditinal image generation experiments are shown in Table 3, 4, 5 (Tab 2 in main paper).
> Additional experiments on Table 6, 7, 8 in x4, x2, x8 setting respectively. Human evaluation for image super-resolution task has shown in **Sec. K2 (Appendix)**.
>
> **Table 2: Comparison of Personalized few-shot finetuning task**
>
> |  Method      | FID score  | MUSIQ score | DINO | CLIP-I | CLIP-T |
> |--------------|-------------|------------|-------|--------|---------|
> | DreamBooth [1]| 111.76 | 68.31 | 0.65 | 0.81 | 0.31 |
> | DreamBooth [1] + LPIPS [3] | 108.23 | 68.39 | 0.65 | 0.80 | 0.32 |
> | DreamBooth [1] + KC loss | **100.08** | **69.78** | **0.68** | **0.84** | **0.34** |
> | Custom Diff.[2] | 84.65 | 70.15 | 0.71 | 0.87 | 0.38 |
> | Custom Diff.[2] + LPIPS [3] | 80.12 | 70.56 | 0.71 | 0.87 | 0.37 |
> | Custom Diff.[2] + KC loss | **75.68** | **72.22** | **0.73** | **0.88** | **0.40** |
>
>
> **Table 3: Comparison of unconditional image generation task (Oxford flowers dataset)**
>
> |  Method      | FID score  | MUSIQ score |
> |--------------|-------------|------------|
> | DDPM [4]| 243.43 | 20.67 |
> | DDPM [4] + LPIPS [3] | 242.62 | 20.80 |
> | DDPM [4] + KC loss | **237.73** | **21.13** |
>
>
> **Table 4: Comparison of unconditional image generation task (Celeb-faces dataset)**
>
> |  Method      | FID score  | MUSIQ score |
> |--------------|-------------|------------|
> | DDPM [4]| 202.67 | 19.07 |
> | DDPM [4] + LPIPS [3] | 201.55 | 19.21 |
> | DDPM [4] + KC loss | **198.23** | **19.52** |
>
>
> **Table 5: Comparison of unconditional image generation task (CelebAHQ dataset)**
>
> |  Method      | FID score  | MUSIQ score |
> |--------------|-------------|------------|
> | DDPM [4]| 199.77 | 46.05 |
> | DDPM [4] + LPIPS [3] | 197.17 | 46.15 |
> | DDPM [4] + KC loss | **190.59** | **46.83** |
>
>
>
> **Table 6: Comparison of image super-resolution (x4) task**
>
> |  Method      | FID score  | PSNR | SSIM | LPIPS | MUSIQ score |
> |--------------|-------------|------------|-------|--------|---------|
> | GD [5]| 121.23 | 18.13 | 0.54 | 0.28 | 57.31 |
> | GD [5] + LPIPS [3] | 119.81 | 18.22 | 0.54 | 0.27 | 57.42 |
> | GD [5] + KC loss | **103.19** | **18.92** | **0.55** | **0.26** | **58.69** |
> | LD [6] | 95.83 | 19.16 | 0.56 | 0.26 | 59.57 |
> | LD [6] + LPIPS [3] | 92.77 | 19.42 | 0.57 | 0.25 | 59.82 |
> | LD [6] + KC loss | **83.34** | **20.25** | **0.58** | **0.22** | **61.20** |
>
>
> **Table 7: Comparison of image super-resolution (x2) task**
>
> |  Method      | FID score  | PSNR | SSIM | LPIPS | MUSIQ score |
> |--------------|-------------|------------|-------|--------|---------|
> | GD [5]| 100.2 | 19.4 | 0.62 | 0.25 | 58.12 |
> | GD [5] + KC loss | **80.9** | **20.2** | **0.66** | **0.20** | **59.91** |
> | LD [6] | 82.45 | 21.2 | 0.64 | 0.24 | 60.23 |
> | LD [6] + KC loss | **70.12** | **22.3** | **0.70** | **0.18** | **62.15** |
>
>
> **Table 8: Comparison of image super-resolution (x8) task**
>
> |  Method      | FID score  | PSNR | SSIM | LPIPS | MUSIQ score |
> |--------------|-------------|------------|-------|--------|---------|
> | GD [5]| 140.3 | 17.5 | 0.52 | 0.32 | 55.26 |
> | GD [5] + KC loss | **125.5** | **18.7** | **0.56** | **0.27** | **57.33** |
> | LD [6] | 103.2 | 18.7 | 0.59 | 0.25 | 58.62 |
> | LD [6] + KC loss | **80.1** | **19.5** | **0.67** | **0.20** | **60.31** |
>
> **References:**
>
> [1] Ruiz, Nataniel, et al. "Dreambooth: Fine tuning text-to-image diffusion models for subject-driven generation." Proceedings of the IEEE/CVF Conference on Computer Vision and Pattern Recognition. 2023.
>
> [2] Nupur Kumari, Bingliang Zhang, Richard Zhang, Eli Shechtman, and Jun-Yan Zhu. Multi-concept customization of text-to-image diffusion. CVPR 2023
>
> [3] Richard Zhang, Phillip Isola, Alexei A Efros, Eli Shechtman, and Oliver Wang. The unreasonable effectiveness of deep features as a perceptual metric. In Proceedings of the IEEE conference on computer vision and pattern recognition, pp. 586–595, 2018.
>
> [4] Ho, Jonathan, Ajay Jain, and Pieter Abbeel. "Denoising diffusion probabilistic models." Advances in neural information processing systems 33 (2020): 6840-6851.
>
> [5] Prafulla Dhariwal and Alexander Nichol. Diffusion models beat gans on image synthesis. Advances in neural information processing systems, 34:8780–8794, 2021.
>
> [6] Tero Karras, Miika Aittala, Timo Aila, and Samuli Laine. Elucidating the design space of diffusionbased generative models. Advances in Neural Information Processing Systems, 35:26565–26577,2022

---

### Meta-Review · Area_Chair_3RN1 · 2023-12-11

**Metareview:**

Summary
This paper aims at improving generation quality and reducing artifacts of diffusion models. Specifically, a kurtosis concentration loss based on statistics of natural images is added to the diffusion model in addition to reconstruction loss in the training. The proposed loss can work with several diffusion-based generation models.

Strengths:
The revisiting kurtosis concentration and adopting it as loss to maintain image naturalness in diffusion models is novel and interesting​​.

Weaknesses:
The contribution of this work is limited, which does not propose any new training strategy or very new loss but instead simply adopting an existing property of natural images as additional loss to diffusion models.
The improvement on generation quality is not obvious according to qualitative comparison, which weakens the contribution of this work.

**Justification For Why Not Higher Score:**

Limited contribution and minor generation quality improvement according to its qualitative results.

**Justification For Why Not Lower Score:**

This work is recommended to be rejected.

---

### Decision · Program_Chairs · 2024-01-16

Reject